# RePL: Pseudo-label Refinement for Semi-supervised LiDAR Semantic Segmentation

## Abstract

Semi-supervised learning for LiDAR semantic segmentation often suffers from error propagation and confirmation bias caused by noisy pseudo-labels. To tackle this chronic issue, we introduce RePL, a novel framework that enhances pseudo-label quality by identifying and correcting potential errors in pseudo-labels through masked reconstruction, along with a dedicated training strategy. We also provide a theoretical analysis demonstrating the condition under which the pseudo-label refinement is beneficial, and empirically confirm that the condition is mild and clearly met by RePL. Extensive evaluations on the nuScenes-lidarseg and SemanticKITTI datasets show that RePL improves pseudo-label quality a lot and, as a result, achieves the state of the art in LiDAR semantic segmentation.

## 1 Introduction

Outdoor LiDAR semantic segmentation, the task of assigning semantic labels to every point in outdoor 3D scenes, plays a crucial role in diverse applications such as autonomous driving (Pendleton et al., 2017; Roriz et al., 2022; Geiger et al., 2012) and robotics (Wang et al., 2024; Serfling et al., 2025). Recent progress in this field has been largely driven by supervised learning on large-scale point cloud datasets (Behley et al., 2019; Fong et al., 2021). However, collecting dense annotations for 3D point clouds is prohibitively costly and time intensive, which limits the scale and class diversity of training data. To alleviate this bottleneck, a large body of research has explored data-efficient training paradigms such as semi-supervised learning (Jiang et al., 2021; Kong et al., 2023; Li et al., 2023; Li & Dong, 2024; Liu et al., 2024; 2025), weakly supervised learning (Liu et al., 2022; Unal et al., 2024), and unsupervised learning (Zhang et al., 2023; Nunes et al., 2023). In this work, we study semi-supervised learning for LiDAR semantic segmentation, where only a subset of 3D scenes for training is manually labeled and the remainder is unlabeled.

At the core of semi-supervised learning lies the challenge of leveraging unlabeled data effectively for training. To this end, existing methods for LiDAR semantic segmentation commonly leverage consistency regularization (Kong et al., 2023; Liu et al., 2024; 2025) and contrastive learning (Jiang et al., 2021; Li & Dong, 2024; Liu et al., 2024). Consistency regularization encourages stable and invariant predictions by enforcing similar outputs under different perturbations of the same input, while contrastive learning promotes feature representations that bring samples of the same pseudo-labels closer and push those of different pseudo-labels farther apart. Although these approaches often yield substantial gains, they have a fundamental vulnerability, a confirmation bias towards erroneous pseudo-labels (Kwon & Kwak, 2022; Yang et al., 2022), since they blindly exploit the model's predictions as pseudo-labels for training the model with unlabeled data; this could cause performance to deteriorate as training progresses.

To handle noisy pseudo-labels, recent studies have proposed confidence-based filtering (Kong et al., 2023; Li et al., 2023; Liu et al., 2025), which discards predictions with low confidence under the assumption that such labels are more error-prone. Loss reweighting methods (Li & Dong, 2024; Liu et al., 2024) take a complementary approach by retaining all pseudo-labeled samples but adjusting their contribution through confidence weighted loss scaling. While both strategies assuage the impact of unreliable labels, they remain post-hoc in nature as they adjust sample utilization only after pseudo-labels have been assigned, rather than improving their quality at the point of generation.

To tackle this issue, we introduce a novel pseudo-label refinement framework, named RePL (**Re**finement of **P**seudo-**L**abels), which is illustrated in Fig. 1. RePL integrates two key com-

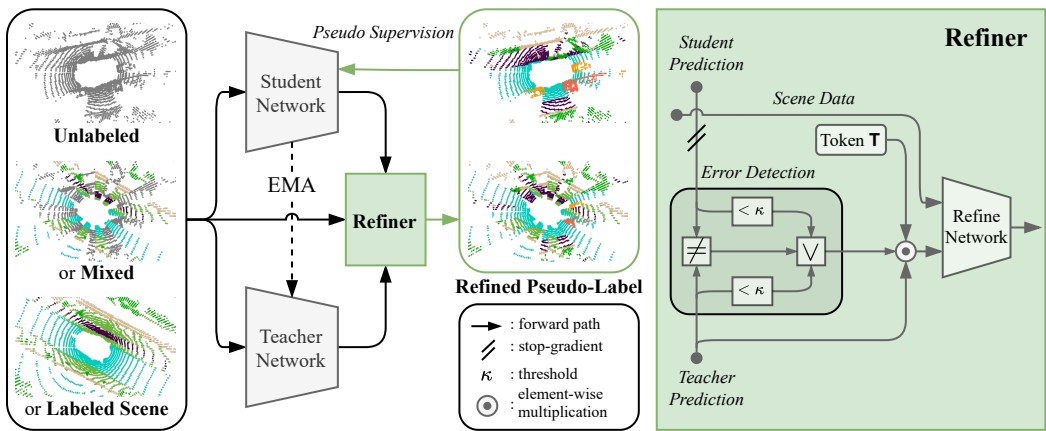

Figure 1: Overview of REPL. The teacher generates predictions for unlabeled LiDAR scenes, which are used as pseudo-labels for the student, and is updated via exponential moving average (EMA) of the student. The pseudo-label refiner detects erroneous pseudo-labels by confidence-based agreement between the teacher and student, and then corrects them through masked reconstruction with learnable tokens. The final refined pseudo-labels combine reliable teacher predictions with reconstructed ones, yielding improved supervision for semi-supervised learning of the student.

ponents: teacher-student networks for LiDAR semantic segmentation (Tarvainen & Valpola, 2017) and a pseudo-label refiner that identifies errors on pseudo-labels and corrects them. The teacher network offers initial predictions for LiDAR scenes, which are in turn used as pseudo-labels for training the student with unlabeled data; the teacher is then updated by exponential moving average (EMA) of the student. The pseudo-label refiner identifies potentially unreliable regions in the pseudo-labels and reconstructs them into cleaner supervisory signals for the student. Specifically, the refiner operates through two stages. First, it estimates potentially erroneous pseudo-labels using a simple confidence-based agreement between teacher and student predictions. Second, it reconstructs these pseudo-labels through a process inspired by masked autoencoders (He et al., 2022); in this step, unreliable areas are replaced with mask tokens, and the refiner reconstructs them to produce refined predictions. Even with a simple error estimation strategy, REPL achieves significant performance improvements, while remaining extensible to more sophisticated error detection methods.

The success of our method heavily depends on the ability of the pseudo-label refiner. However, training the refiner in the semi-supervised learning setting is challenging due to the scarce supervision: the supervision derived from the discrepancies between predicted and ground-truth labels is available only for a small subset of data. We address this issue in two ways. First, during training the refiner, we apply random masking to make its reconstruction more challenging. This forces the refiner to develop a better contextual understanding rather than simply memorizing patterns. Second, we mix 3D scenes, augmenting labeled scenes with unlabeled ones before feeding them to the segmentation networks. This produces strongly augmented segmentation predictions with higher and diverse error rates, and enables the refiner to partially experience prediction errors for unlabeled images. We also provide a theoretical analysis demonstrating the condition under which the refinement is beneficial, and empirically confirm that the condition is mild and clearly met by REPL.

Following the established practice (Kong et al., 2023; Liu et al., 2024; 2025), we evaluated our method on the nuScenes-lidarseg (Fong et al., 2021) and SemanticKITTI (Behley et al., 2019) benchmarks while varying the ratio of labeled data, where it demonstrated significant performance improvements over the supervised learning baseline and outperformed latest methods for semi-supervised learning. In summary, our contribution is three-fold:

- We propose a semi-supervised LiDAR semantic segmentation framework, dubbed REPL, that refines pseudo-labels via error estimation and masked reconstruction.

- We provide a theoretical analysis establishing the condition under which pseudo-label refinement improves upon teacher-only baseline.

- Our method achieved the state of the art on two public benchmarks.

## 2 RELATED WORK

**LiDAR Semantic Segmentation** LiDAR semantic segmentation assigns semantic labels to each point in large-scale point clouds (Zhu et al., 2022; Milioto et al., 2019). Early work transformed raw 3D scans into 2D representations such as range images or bird's-eye-view maps to reuse established 2D convolutional backbones. RangeNet++ (Milioto et al., 2019) and SalsaNext (Cortinhal et al., 2020) demonstrated that simple projection can yield competitive results. Another research stream discretized 3D space into regular or cylindrical grids, as seen in PolarNet (Zhang et al., 2020), Cylinder3D (Zhu et al., 2022), and sparse convolutional frameworks like MinkowskiNet (Choy et al., 2019). Meanwhile, methods directly processing raw points gained traction, from PointNet (Qi et al., 2016) to RandLA-Net (Hu et al., 2020) and stratified architectures (Jiang et al., 2019; Lai et al., 2022), capturing both local details and long-range context. Despite performance gains, these methods require dense manual annotations, which are costly and not scalable.

**Semi-supervised LiDAR Semantic Segmentation** To address this annotation bottleneck, semi-supervised learning leverages limited labeled data along with abundant unlabeled point clouds (Kong et al., 2023; Liu et al., 2024). Earlier work improved pseudo-label reliability: GPC (Jiang et al., 2021) used confidence thresholds to reduce error propagation, LaserMix (Kong et al., 2023) exploited spatial priors of LiDAR beams, enforcing prediction consistency through beam mixing across scans, and Lim3D (Li et al., 2023) employed a memory bank for contrastive learning to alleviate class imbalance. Recent methods introduced richer constraints: DDSemi (Li & Dong, 2024) employed density-guided contrastive learning with dual-space hardness sampling for sparse regions, AIScene (Liu et al., 2025) addressed intra-scene inconsistency through patch-based mixing at scene and instance levels, and IT2 (Liu et al., 2024) introduced consistency learning across peer LiDAR representations, treating representation differences as perturbations. These approaches remain post-hoc, adjusting pseudo-label usage rather than improving their intrinsic quality. REPL directly enhances pseudo-labels by correcting erroneous pseudo-labels, providing improved supervision.

## 3 METHOD

The main challenge in semi-supervised learning is the confirmation bias of pseudo-labels, which has often been handled by post-hoc strategies such as confidence filtering or loss reweighting. In this work, we explore a different direction: instead of discarding unreliable pseudo-labels, we aim to improve their quality through refinement.

We propose REPL, a refinement-based semi-supervised learning framework for LiDAR semantic segmentation. The framework consists of two modules: teacher–student segmentation networks and a pseudo-label refiner. The teacher network generates pseudo-labels for unlabeled data and is updated by exponential moving average of the student parameters, while the student is trained with both labeled and pseudo-labeled data. To handle unreliable pseudo-labels, the refiner identifies uncertain voxels and corrects their pseudo-labels through masked reconstruction. The final pseudo-labels combine reliable teacher predictions with the refined output on uncertain regions.

The remainder of this section first elaborates on three training steps of REPL: (1) training the student network using the standard segmentation objectives on labeled data (Section 3.2), (2) training the pseudo-label refiner on both labeled and unlabeled data (Section 3.3), and (3) semi-supervised learning of the student network with the pseudo-labels improved by the refiner (Section 3.4). Then a theoretical analysis is presented to establish when the refinement is beneficial and to validate that REPL operates within this regime (Section 3.5).

### 3.1 PRELIMINARIES

We consider a segmentation network $f$ trained on a small labeled dataset and a large unlabeled dataset. Each point cloud is voxelized into a regular grid $X_i \in \mathbb{R}^{C \times H \times W \times L}$, where $C$ denotes the number of feature channels and $H \times W \times L$ is the grid size. The labels are converted into voxel-wise one-hot tensors $Y_i \in \{0, 1\}^{K \times H \times W \times L}$, where $K$ is the number of classes. We denote the voxelized labeled and unlabeled datasets as $D_L = \{(X_i, Y_i)\}_{i=1}^{N_l}$ and $D_U = \{X_j\}_{j=1}^{N_u}$, respectively.

## 3.2 SUPERVISED LEARNING WITH LABELED DATA

Following prior work (Zhu et al., 2022; Kong et al., 2023; Li et al., 2023; Liu et al., 2024), we train the student network $f(\cdot)$ on labeled data with two complementary segmentation losses: cross-entropy for voxel-wise classification and Lovász-Softmax (Berman et al., 2018) for direct IoU optimization. We denote the set of voxel grid indices as $\Omega = \{1, \ldots, H\} \times \{1, \ldots, W\} \times \{1, \ldots, L\}$. For input $X_i$, the network outputs predictions $P_i = f(X_i) \in \mathbb{R}^{K \times H \times W \times L}$, where $[P_i]_{k,\omega}$ denotes the probability of class $k$ at voxel $\omega \in \Omega$. The ground-truth label is denoted by $[Y_i]_{k,\omega} \in \{0, 1\}$.

The voxel-wise cross-entropy loss is defined as

$$\mathcal{L}_{\mathrm{ce}}(P_i, Y_i) = -\frac{1}{|\Omega|} \sum_{\omega \in \Omega} \sum_{k=1}^{K} [Y_i]_{k,\omega} \, \log [P_i]_{k,\omega}. \tag{1}$$

For Lovász-Softmax, we first define the one-versus-rest error for each class $k$:

$$\mathbf{e}_i^{(k)}(\omega) = (1 - [P_i]_{k,\omega}) \cdot \mathbf{1}_{\{[Y_i]_{k,\omega}=1\}} + [P_i]_{k,\omega} \cdot \mathbf{1}_{\{[Y_i]_{k,\omega}\neq 1\}}. \tag{2}$$

The multiclass Lovász extension is then given by:

$$\mathcal{L}_{\mathrm{ls}}(P_i, Y_i) = \frac{1}{|C_i'|} \sum_{k \in C_i'} \overline{\Delta}_{\mathrm{Jacc}}\big(\mathbf{e}_i^{(k)}\big), \tag{3}$$

where $C_i' = \{k \mid \sum_{\omega \in \Omega}[Y_i]_{k,\omega} > 0\}$ and $\overline{\Delta}_{\mathrm{Jacc}}$ denotes the Lovász extension (Berman et al., 2018) of the Jaccard loss. The complete supervised learning objective combines both terms:

$$\mathcal{L}_{\mathrm{ssup}} = \frac{1}{N_l} \sum_{i=1}^{N_l} \Big\{ \mathcal{L}_{\mathrm{ce}}(P_i, Y_i) + \lambda_{\mathrm{ls}} \, \mathcal{L}_{\mathrm{ls}}(P_i, Y_i) \Big\}. \tag{4}$$

## 3.3 TRAINING PSEUDO-LABEL REFINER

We employ the teacher-student framework where the teacher network, $f^\tau(\cdot)$, generates pseudo-labels for unlabeled data, $Q_j = f^\tau(X_j)$, and is updated by exponential moving average of student parameters (Tarvainen & Valpola, 2017). The pseudo-label refiner network $g(\cdot)$ is designed to identify and reconstruct unreliable teacher's predictions given as pseudo-labels for unlabeled scenes.

**Unreliable Voxel Identification.** Voxels with unreliable pseudo-labels are identified by the agreement between the student's and teacher's predictions along with their confidence levels. For each voxel, we compute the confidence score as the maximum prediction probability across all classes for each of the two networks. We also establish adaptive confidence thresholds using the $(100 - \kappa)$-th percentile of all confidence scores within each scene, where $\kappa$ controls the strictness of the confidence requirement. A voxel is considered reliable only if the following conditions hold: (1) the student and teacher predict the same class, (2) the student's confidence exceeds its adaptive threshold, and (3) the teacher's confidence exceeds its adaptive threshold. All other voxels are treated as unreliable and marked for refinement. The error candidate mask is then given by $M = \mathbf{1} - \mathbf{1}_{\mathrm{rel}} \in \{0, 1\}^{H \times W \times L}$, where $\mathbf{1}_{\mathrm{rel}}$ denotes the reliable mask indicating voxels that satisfy all the three conditions above. To prevent the refiner from overfitting to error-prone regions and develop a better contextual understanding rather than simply memorizing patterns, we also apply additional random masking $R \sim \mathrm{Bernoulli}(\sigma)$ and define the final mask as $\tilde{M} = M \vee R$.

**Masked Reconstruction.** Once unreliable voxels are identified through the error candidate mask, the next step is to correct their predictions through a masked reconstruction process (He et al., 2022). The core idea is to mask out these uncertain predictions and train the refiner to reconstruct more accurate predictions for them. Specifically, unreliable predictions are replaced with a learnable mask token $T$: $\bar{Q} = (\mathbf{1} - \tilde{M}) \odot Q + \tilde{M} \odot T$, where $\odot$ denotes element-wise multiplication. Then the refiner takes channel-wise concatenated $(X, \bar{Q})$ as input and outputs refined predictions $\hat{Q} = g(X, \bar{Q})$.

**Training on Labeled Data.** On labeled data, the refiner is trained to reconstruct ground-truth labels of the masked predictions. The loss for training the refiner on labeled data, $\mathcal{L}_{\mathrm{rsup}}$, is the same as the supervised learning objective $\mathcal{L}_{\mathrm{ssup}}$ in Eq. (4), except that it is applied to only the masked regions (*i.e.*, $\forall \omega \in \Omega$ s.t. $[\tilde{M}_i]_\omega = 1$) of the refined predictions $\hat{Q}_i$, instead of $P_i$.

**Training on Unlabeled Data.** On unlabeled data, the refiner is further trained with a negative learning signal (Kim et al., 2019). Rather than enforcing hard pseudo-labels, we suppress predictions on implausible classes by taking the teacher's top-$k$ predictions as plausible candidates and encouraging the refiner to assign low probability to the remaining classes. This negative learning strategy offers reliable supervisory signals even when pseudo-supervision may not be sufficiently accurate on unlabeled data. Formally, the negative learning loss is defined as the average cross-entropy penalty over unlabeled scenes in $D_U$:

$$\mathcal{L}_{\text{runl}} = \frac{1}{N_u} \sum_{j=1}^{N_u} \frac{1}{|\Omega|} \sum_{\omega \in \Omega} \frac{1}{|\mathcal{N}_j(\omega)|} \sum_{k \in \mathcal{N}_j(\omega)} \left\{ -\log(1 - [\hat{Q}_j]_{k,\omega}) \right\}, \tag{5}$$

where $\mathcal{N}_j(\omega)$ denotes the set of implausible classes for voxel $\omega$.

**Training on Mix of Labeled and Unlabeled Data.** Training the refiner benefits from challenging prediction errors of the teacher, but the limited labeled data restrict the diversity of such errors. To strengthen the supervision, we mix labeled and unlabeled scenes so that the refiner reconstructs labels under richer variability in distance, geometry, and density. This produces augmented predictions with higher error rates, and allows the refiner to partially experience errors on unlabeled data. We adopt LaserMix (Kong et al., 2023) with a single inclination plane to fuse labeled and unlabeled scans at ratio $r \in (0, 1)$. Let a mix of labeled and unlabeled scenes be indexed by $m \in \{1, \ldots, N_m\}$. Given a labeled scene $(X_i, Y_i)$ and an unlabeled one $X_j$ for the $m$-th mix, LaserMix generates a selector mask $S_m \in \{0, 1\}^{H \times W \times L}$ and produces the mixed input and output $(X_m, Y_m) = (S_m \odot X_i + (\mathbf{1} - S_m) \odot X_j, S_m \odot Y_i)$. We then compute student and teacher predictions on $X_m$, $P_m = f(X_m)$ and $Q_m = f^\tau(X_m)$, respectively. Following the unreliable prediction identification procedure in Section 3.3, we construct the error mask $\tilde{M}_m$ restricted to the labeled prediction. The supervised learning losses are then applied to voxels of the labeled scene marked as unreliable, *i.e.*, $\forall \omega \in \Omega$ s.t. $[S_m]_\omega = [\tilde{M}_m]_\omega = 1$:

$$\mathcal{L}_{\text{rmix}} = \frac{1}{N_m} \sum_{m=1}^{N_m} \left\{ \mathcal{L}_{\text{ce}}(\hat{Q}_m, Y_m) + \lambda_{\text{ls}} \mathcal{L}_{\text{ls}}(\hat{Q}_m, Y_m) \right\}. \tag{6}$$

**Total Training Objective.** Each iteration optimizes the pseudo-label refiner with the summation of the three losses, $\mathcal{L}_{\text{rsup}} + \mathcal{L}_{\text{runl}} + \mathcal{L}_{\text{rmix}}$, without balancing hyper-parameters.

### 3.4 SEMI-SUPERVISED LEARNING WITH PSEUDO-LABEL REFINER

The student network leverages the refined pseudo-labels from the refiner to improve the quality of supervision when learning with unlabeled data.

**Pseudo-label Refinement.** For an unlabeled input $X_j \in D_U$, we obtain predictions from both student and teacher $(P_j, Q_j)$ and construct the error mask $M_j$ as before but without random masking. The refined pseudo-label $\tilde{Y}_j$ is then generated voxel-wise: reliable voxels follow the teacher's prediction, while unreliable ones are replaced with the refiner's output. Specifically, we form masked predictions $\bar{Q}_j = (\mathbf{1} - M_j) \odot Q_j + M_j \odot T$ and obtain refined predictions $\hat{Q}_j = g(X_j, \bar{Q}_j)$, which are combined with teacher predictions on reliable voxels to produce $\tilde{Y}_j$.

**Semi-supervised Learning of Student.** The student network is trained with three objectives, the supervised learning objective applied to labeled data (*i.e.*, $\mathcal{L}_{\text{ssup}}$ in Section 3.2), and two additional objectives for semi-supervised learning with unlabeled data. For the segmentation losses on unlabeled data, since the refined pseudo-labels are used directly without additional filtering, we adopt the symmetric cross-entropy (Wang et al., 2019) instead of the standard cross-entropy to ensure more stable training. The symmetric formulation is more robust to potential noise in pseudo-labels as it penalizes over-confident predictions and provides regularization through bidirectional loss computation. Together with the Lovász-Softmax loss, this objective is applied to all voxels using the refined pseudo-labels:

$$\mathcal{L}_{\text{sunl}} = \frac{1}{N_u} \sum_{j=1}^{N_u} \left\{ \frac{1}{2} \left( \mathcal{L}_{\text{ce}}(P_j, \tilde{Y}_j) + \mathcal{L}_{\text{ce}}(\tilde{Y}_j, P_j) \right) + \lambda_{\text{ls}} \mathcal{L}_{\text{ls}}(P_j, \tilde{Y}_j) \right\}. \tag{7}$$

In addition to training with pseudo-labels, we augment training by mixing a labeled scene $(X_i, Y_i)$ with an unlabeled scene $X_j$ paired with pseudo-labels $\tilde{Y}_j$. We employ the LaserMix operator with a single inclination plane. This strategy combines clean supervision from labeled data with broader coverage from pseudo-labels, exposing the student to reliable signals and diverse structures. Given the selector mask $S_m$ by LaserMix, the mixed input and target are defined by combining ground-truth labels in the labeled region and refined pseudo-labels in the unlabeled region:

$$(X_m, \tilde{Y}_m) = \left( S_m \odot X_i + (\mathbf{1} - S_m) \odot X_j, \ S_m \odot Y_i + (\mathbf{1} - S_m) \odot \tilde{Y}_j \right). \tag{8}$$

For the mixed input $X_m$, the student network produces predictions $P_m = f(X_m)$. The loss for the mixed sample is then computed by

$$\mathcal{L}_{\text{smix}} = \frac{1}{N_m} \sum_{j=1}^{N_m} \left\{ \tfrac{1}{2} \left( \mathcal{L}_{\text{ce}}(P_m, \tilde{Y}_m) + \mathcal{L}_{\text{ce}}(\tilde{Y}_m, P_m) \right) + \lambda_{\text{ls}} \mathcal{L}_{\text{ls}}(P_m, \tilde{Y}_m) \right\}. \tag{9}$$

**Total Training Objective.** In each iteration, the student network is optimized with the summation of the three losses, $\mathcal{L}_{\text{ssup}} + \mathcal{L}_{\text{sunl}} + \mathcal{L}_{\text{smix}}$, with no balancing hyper-parameter. The student network is optimized jointly with the pseudo-label refiner. We stop gradients between their optimization paths to prevent interference.

### 3.5 THEORETICAL ANALYSIS

This section rigorously analyzes if the pseudo-label refinement is truly helpful. We first show whether the pseudo-label refinement is easier than generating high-quality pseudo-labels from scratch in Proposition 1.

**Proposition 1** (Task Difficulty). *Consider two segmentation tasks, the original task $Z : X \to Y$ and the refinement task $Z' : (X, T) \to Y$, where $X$ denotes input 3D LiDAR point data, $Y$ denotes segmentation labels, and $T$ represents additional information such as teacher predictions $f(X)$. The difficulties of the two tasks $D(Z)$ and $D(Z')$ hold the following inequality:*

$$D(Z') = H(Y \mid X, T) \leq H(Y \mid X) = D(Z). \tag{10}$$

This result implies that the refinement may have potential for improving pseudo-label quality. Next, we derive a practical condition required for net performance gains in Proposition 2.

**Proposition 2** (Improvement Condition). *For the $j$-th scene, an error-candidate mask $M_j$ divides the voxel grid into an unreliable region $E_j$ and a reliable region $C_j$. Let $\pi_j$ denote the precision of this mask, which is the fraction of voxels misclassified by the teacher in $E_j$. The refiner operates on $E_j$, where $q_j$ and $r_j$ denote the rates at which it corrects misclassifications and incorrectly changes correct predictions, respectively. Then, if $q_j + r_j > 0$, the refinement improves the accuracy for scene $j$ if and only if*

$$\zeta_j := \pi_j - \frac{r_j}{q_j + r_j} > 0. \tag{11}$$

This proposition characterizes the trade-off between error correction and error introduction, and its conclusion is a condition that is mild and easily satisfied by REPL even with the simple error estimation method in Section 3.4, as will be demonstrated below. The proofs for the two propositions are presented in Appendices A.1 and A.2.

**Empirical Analysis on the Improvement Condition.** To examine the practical implication of the condition in Eq. (11), we analyze the relationship between the averaged correction rate $q$ and the averaged error introduction rate $r$ using experimental results on the SemanticKITTI dataset. We consider two scenarios: labeling 1% of training data yielding $\pi = 0.917$, and labeling 50% achieving $\pi = 0.983$. Figure 2 visualizes combinations of $q$ and $r$ where the refinement yields a net benefit ($\zeta > 0$) versus those where it does not ($\zeta \leq 0$). The results suggest that the refinement remains beneficial across a broad range of $q$ and $r$. For instance, in the case of $\pi = 0.917$, the refinement remains beneficial as long as the error introduction rate stays below approximately *eleven times* the correction rate ($r < 11.05 \cdot q$), allowing the refinement to be effective across a broad range of performance levels. In both experimental cases, REPL falls within the benefit regions ($\zeta > 0$), demonstrating that it satisfies the theoretical condition for the improvement of pseudo-label quality despite employing a simple error estimation strategy.

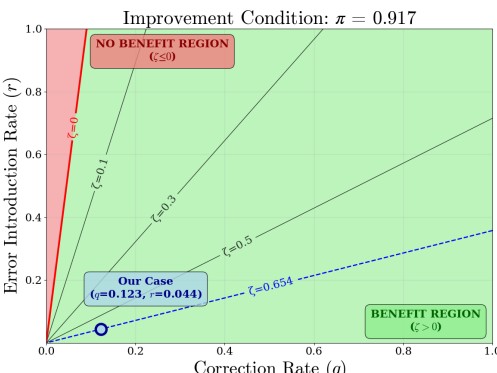 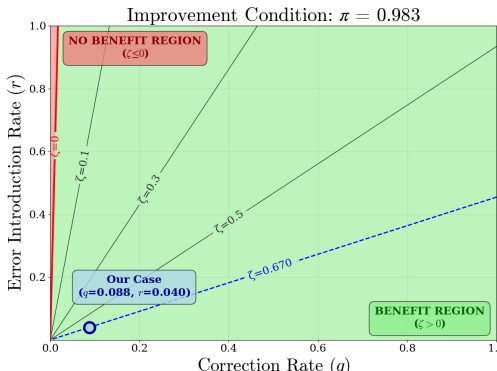

Figure 2: Visualization of the improvement condition from Eq. (11) on the SemanticKITTI dataset given $\pi$. The two values of $\pi$ were derived during the actual experiments on the dataset: 1% labeled data ($\pi = 0.917$) and 50% labeled data ($\pi = 0.983$). Green areas indicate combinations of $q$ and $r$ where the refinement yields a net benefit ($\zeta > 0$), while red areas show such combinations leading to detriment ($\zeta \leq 0$). The results suggest that the refinement remains beneficial across a broad range of $q$ and $r$, and that REPL clearly helps improve the quality of pseudo-labels.

## 4 EXPERIMENTS

### 4.1 EXPERIMENTAL SETUP

**Datasets.** We trained and evaluated our model on two outdoor LiDAR semantic segmentation benchmarks: nuScenes-lidarseg (Fong et al., 2021) and SemanticKITTI (Behley et al., 2019). nuScenes-lidarseg is a large-scale outdoor LiDAR dataset with point-wise annotations for 16 classes. We adopted the official split of 700 sequences for training and 150 for validation, which resulted in 28,130 training and 6,019 validation point cloud scenes. SemanticKITTI is a LiDAR segmentation benchmark with 19 classes. It consists of 22 sequences, of which 10 were used for training, and 1 for validation, yielding 19,130 training and 4,071 validation scenes.

**Network Architecture.** Following previous work (Kong et al., 2023; Liu et al., 2024; 2025), we used Cylinder3D (Zhu et al., 2022) for both the segmentation models and pseudo-label refiner, fixing the intermediate layer at 16 dimensions as specified by Kong et al. (2023) and Liu et al. (2024).

**Implementation Details.** Our method was implemented in PyTorch (Paszke et al., 2017), and trained on 8 NVIDIA RTX 6000 Ada GPUs with AdamW (Loshchilov & Hutter, 2019) and weight decay of 1e-3. The batch size was 8 on nuScenes-lidarseg and 4 on SemanticKITTI. The learning rate was 5e-3 with cosine annealing for both the segmentation network and pseudo-label refiner. The weight update ratio $\alpha$ was 0.994. Following Liu et al. (2024), the loss coefficient $\lambda_{ls}$ was set to 3.0. We set the confidence percentile $\kappa$ to 40%, $k$ of top-$k$ classes for negative learning to 3, random masking probability $\sigma$ to 0.15, and mixed data at a mixing ratio $r$ of 0.7 in the selector mask $S$. Note that all hyper-parameters except the batch size were set identically across the two benchmarks.

### 4.2 COMPARISON WITH STATE OF THE ART

REPL was compared with latest methods using Cylinder3D as the backbone, namely AIScene (Liu et al., 2025), IT2 (Liu et al., 2024), FrustrumMix (Xu et al., 2025), and conventional semi-supervised methods (Tarvainen & Valpola, 2017; Zou et al., 2018; Chen et al., 2021; Kong et al., 2023; 2025). For a more comprehensive comparison, we further evaluated against Seal (Liu et al., 2023), Super-Flow (Xu et al., 2024), and SLidR (Sautier et al., 2022), which leverage external sources or additional representation learning, and Lim3D (Li et al., 2023), which uses a distinct backbone based on Cylinder3D. Table 1 summarizes the results on the validation sets of nuScenes-lidarseg (Fong et al., 2021) and SemanticKITTI (Behley et al., 2019) with various ratio of labeled data: 1%, 10%, 20%, and 50%. *Sup-only* denotes the baseline performance of training only with labeled data. On nuScenes-lidarseg, REPL outperformed all competing methods. Compared to IT2, the second-best, it achieved an average gain of +2.0 in mIoU. On SemanticKITTI, REPL also showed strong results, achieving the best performance at 1% and 50%, and the second-best at 10% and 20%. Overall,

Table 1: Comparison of different semi-supervised learning methods on nuScenes-lidarseg and SemanticKITTI while varying the ratio of labeled data. The best results in each column are shown in bold, and the second best are underlined. Backbones marked with an asterisk (*) indicate that additional representation learning or knowledge distillation from external sources has been applied.

| Method | Backbone | nuScenes-lidarseg (Fong et al., 2021) | | | | | SemanticKITTI (Behley et al., 2019) | | | | |
|---|---|---|---|---|---|---|---|---|---|---|---|
| | | 1% | 10% | 20% | 50% | Avg. | 1% | 10% | 20% | 50% | Avg. |
| *Sup-only* | Cylinder3D | 50.9 | 65.9 | 66.6 | 71.2 | 63.7 | 45.4 | 56.1 | 57.8 | 58.7 | 54.5 |
| Seal (Liu et al., 2023) | MinkUNet* | 45.8 | 63.0 | - | - | - | 46.6 | - | - | - | - |
| SuperFlow (Xu et al., 2024) | MinkUNet* | 48.1 | 64.5 | - | - | - | 48.4 | - | - | - | - |
| SLidR (Sautier et al., 2022) | Cylinder3D* | 39.0 | 58.8 | - | - | - | 44.6 | - | - | - | - |
| Lim3D (Li et al., 2023) | LiM3D | - | - | - | - | - | 58.4 | 59.5 | 63.1 | 63.6 | 61.2 |
| MT (Tarvainen & Valpola, 2017) | Cylinder3D | 51.6 | 66.0 | 67.1 | 71.7 | 64.1 | 45.4 | 57.1 | 59.2 | 60.0 | 55.4 |
| CBST (Zou et al., 2018) | Cylinder3D | 53.0 | 66.5 | 69.6 | 71.6 | 65.2 | 48.8 | 58.3 | 59.4 | 59.7 | 56.6 |
| CPS (Chen et al., 2021) | Cylinder3D | 52.9 | 66.3 | 70.0 | 72.5 | 65.4 | 46.7 | 58.7 | 59.6 | 60.5 | 56.4 |
| LaserMix (Kong et al., 2023) | Cylinder3D | 55.3 | 69.9 | 71.8 | 73.2 | 67.6 | 50.6 | 60.0 | 61.9 | 62.3 | 58.7 |
| IT2 (Liu et al., 2024) | Cylinder3D | 57.5 | 72.1 | 73.5 | 74.1 | 69.3 | 52.0 | 61.4 | 62.1 | 62.5 | 59.5 |
| AIScene (Liu et al., 2025) | Cylinder3D | 56.6 | 70.2 | 72.8 | 73.9 | 68.4 | 54.5 | 63.3 | 63.7 | 64.3 | 61.5 |
| FrustrumMix (Xu et al., 2025) | Cylinder3D | 60.0 | 70.0 | 72.6 | 74.1 | 69.2 | 55.7 | 62.5 | 63.0 | 64.9 | 61.5 |
| LaserMix++ (Kong et al., 2025) | Cylinder3D | 58.5 | 71.1 | 72.8 | 74.0 | 69.1 | 56.2 | 62.3 | 62.9 | 63.4 | 61.2 |
| **REPL (Ours)** | Cylinder3D | **60.0** | **74.4** | **75.0** | **75.8** | **71.3** | 54.7 | 62.5 | 63.2 | **65.9** | **61.6** |

Table 2: Impact of the losses for the pseudo-label refiner. Each row shows the average improvement condition $\zeta$ and mean IoU when training with different subsets of the losses.

| $\mathcal{L}_{\text{rsup}}$ | $\mathcal{L}_{\text{runl}}$ | $\mathcal{L}_{\text{rmix}}$ | $\zeta$ | mIoU |
|---|---|---|---|---|
| | | | - | 50.9 |
| ✓ | | | 0.327 | 57.2 |
| ✓ | ✓ | | 0.353 | 58.7 |
| ✓ | ✓ | ✓ | 0.430 | 60.0 |

Table 3: Impact of the losses for learning the LiDAR semantic segmentation network. In $\mathcal{L}_{\text{sunl}}$, ▲ denotes the omission of the symmetric cross-entropy.

| $\mathcal{L}_{\text{ssup}}$ | $\mathcal{L}_{\text{sunl}}$ | $\mathcal{L}_{\text{smix}}$ | mIoU |
|---|---|---|---|
| ✓ | | | 50.9 |
| ✓ | ✓ | | 58.1 |
| ✓ | ▲ | ✓ | 58.0 |
| ✓ | ✓ | ✓ | 60.0 |

Table 4: Sensitivity to the quality of the error candidate mask in LiDAR semantic segmentation accuracy. Different error mask generation strategies were compared at inference time.

| Setting | Baseline | Random | | | Oracle | Ours |
|---|---|---|---|---|---|---|
| | | 25% | 50% | 75% | | |
| mIoU | 57.0 | 57.6 | 58.2 | 58.7 | 67.3 | 60.0 |

Table 5: Impact of the random masking strategy for training the pseudo-label refiner in the segmentation quality of the final model.

| Setting | mIoU |
|---|---|
| w/o Random Masking | 57.7 |
| w/ Random Masking | 60.0 |

REPL achieved the highest average mIoU. Figure 3 qualitatively compares pseudo-labels before and after the refinement by REPL at the end of training on the unlabeled data of nuScenes-lidarseg.

### 4.3 IN-DEPTH ANALYSIS

This section studies the contribution of each component of REPL, and investigates the aspects of the pseudo-label refinement in details. All experiments were conducted on the validation set, except for the pseudo-label refinement analysis, which used the unlabeled training data.

**Ablation Study on Loss Components.** We assessed the contribution of individual loss terms by incrementally adding them for the refiner and segmentation network. For the refiner (Table 2), the supervised-only baseline yielded 50.9 mIoU. Adding $\mathcal{L}_{\text{rsup}}$ improved performance to 57.2 mIoU, and including $\mathcal{L}_{\text{runl}}$ further raised it to 58.7 mIoU. With all three objectives, including $\mathcal{L}_{\text{rmix}}$, accuracy reached 60.0 mIoU, confirming their complementary effect. The averaged improvement condition $\zeta$ also consistently increased as each loss component is added. For the segmentation network (Table 3), the supervised-only baseline also scored 50.9 mIoU. Adding the semi-supervised loss, $\mathcal{L}_{\text{sunl}}$, improved performance to 58.1 mIoU, while its variant without symmetric cross-entropy (▲) with $\mathcal{L}_{\text{smix}}$ gave 58.0 mIoU. Using all three training objectives yielded the best result of 60.0 mIoU, highlighting the benefit of jointly optimizing supervised learning on labeled data, semi-supervised learning with refined pseudo-labels, and mixed scene training.

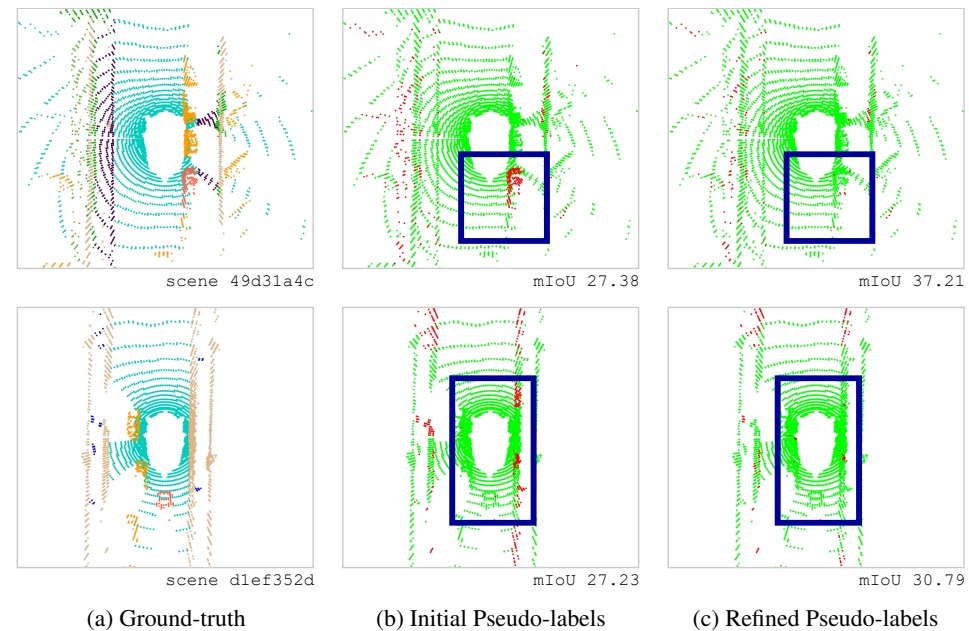

|  |  |  |
| :---: | :---: | :---: |
| (a) Ground-truth | (b) Initial Pseudo-labels | (c) Refined Pseudo-labels |

Figure 3: Qualitative results of refined pseudo-labels and their initial predictions on the unlabeled set of nuScenes-lidarseg, at the end of training. Correct and incorrect predictions are shown in green and red, respectively. The model was trained with a 1% label ratio.

Table 6: Sensitivity analysis of a hyper-parameter $\kappa$.

| $\kappa$ | mIoU |
| :---: | :---: |
| 0.2 | 55.1 |
| 0.4 | 60.0 |
| 0.6 | 58.4 |

Table 7: Computational cost analysis on nuScenes-lidarseg.

| Method | Latency (s) | Memory (MB) | mIoU |
| :--- | :---: | :---: | :---: |
| Baseline | 0.43 | 1231 | 50.9 |
| Baseline + Refiner | 0.68 | 1627 | 60.0 |
| $\Delta$ | +0.25 | +396 | +9.1 |

**Sensitivity to the Quality of Error Candidate Mask.** We analyzed how the quality of the error candidate mask influences inference performance by replacing our heuristic error mask with different alternatives on the validation set. As shown in Table 4, random masks yielded modest improvements over the baseline (no refinement) of the teacher. Our heuristic error mask provided a clear gain, while an oracle error mask derived from ground-truth labels further improved performance to 67.3 mIoU. These results indicate that even a simple heuristic achieves competitive improvements, with more accurate error mask offering substantial room for further gains.

**Impact of the Random Masking Strategy.** We investigated whether training with random masking improves performance on the validation set. As shown in Table 5, incorporating random masking yielded higher performance (60.0 mIoU) compared to training without it (57.7 mIoU). This indicates that random masking serves as a regularizer, helping the network handle erroneous predictions more effectively and improving performance during inference.

**Analysis on Computational Cost.** To quantify the additional overhead by the refiner, we measured the latency and memory usage during inference on the validation set using a single batch. As shown in Table 7, the refiner adds approximately 0.25 seconds of latency and 396 MB of memory, while providing a substantial improvement of +9.1 mIoU from the supervised-only baseline. These results demonstrate that the added computational cost is moderate relative to the significant accuracy gains.

**Ablation Study on Unreliable Voxel Identification.** We analyzed the sensitivity of an unreliable voxel identification in REPL on the validation set. As shown in Table 6, the confidence percentile $\kappa = 0.4$ yields the best performance at 60.0 mIoU, while $\kappa = 0.2$ and $\kappa = 0.6$ result in suboptimal performance at 55.1 and 58.4 mIoU, respectively.

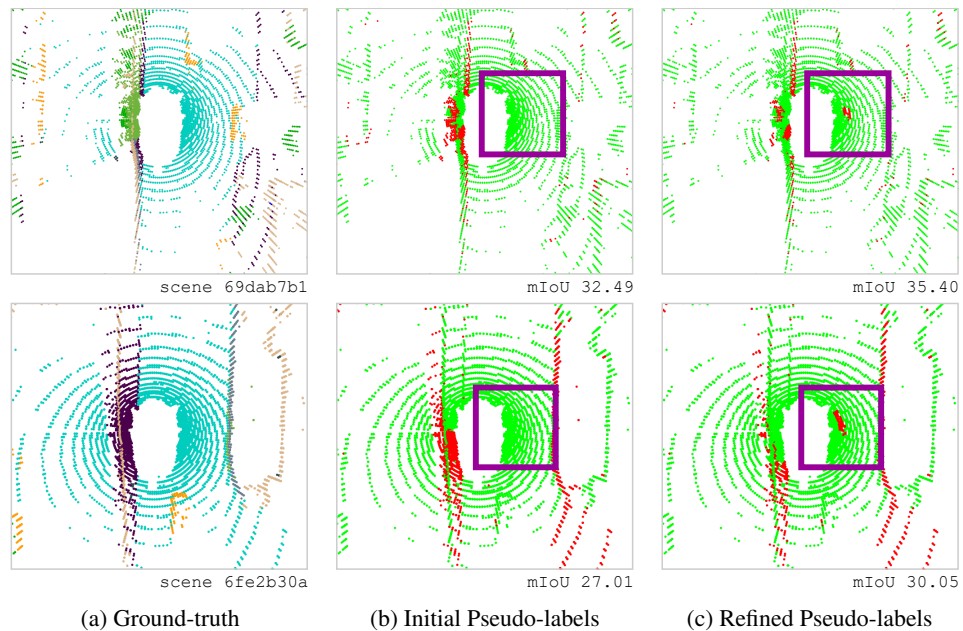

|                     |                     |                     |
|:-------------------:|:-------------------:|:-------------------:|
| scene 69dab7b1      | mIoU 32.49          | mIoU 35.40          |
| scene 6fe2b30a      | mIoU 27.01          | mIoU 30.05          |
| (a) Ground-truth    | (b) Initial Pseudo-labels | (c) Refined Pseudo-labels |

Figure 4: Failure cases on the unlabeled set of nuScenes-lidarseg at the end of training with a 1% label ratio. Correct and incorrect predictions are shown in green and red, respectively.

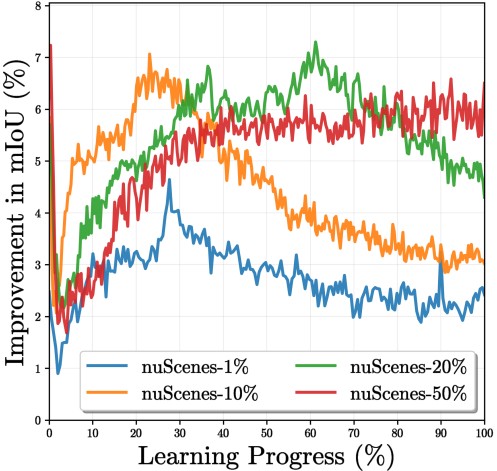

Figure 5: Pseudo-label quality improvement by the refiner during training on nuScenes-lidarseg.

**Analysis on Pseudo-label Quality Improvement throughout Training.** We report the trend of pseudo-label quality improvement throughout training for different labeled data ratios (1%, 10%, 20%, 50%) on the unlabeled data of nuScenes-lidarseg in Figure 5. During early stage of training, the improvement was relatively low across all ratios as the refiner learns error correction from scratch. As training progresses, the improvement increased as the refiner learns to correct errors more effectively. However, the improvement gradually declined in later stages as the segmentation network itself becomes accurate, leaving less room for the refiner to provide meaningful corrections to the already high-quality predictions. REPL showed effectiveness across all labeled data ratios, with better performance and scalability at higher ratios.

**Analysis on Failure Cases.** Despite overall improvements, REPL occasionally introduces errors by over-correcting initially accurate predictions. Figure 4 shows representative failure cases (purple boxes). Nevertheless, the mIoU gain indicates that successful corrections outweigh these localized failures, leading to overall enhancement of the pseudo-labels.

## 5 CONCLUSION

We presented REPL, a semi-supervised learning framework for LiDAR semantic segmentation that refines pseudo-labels through a two-stage mechanism of error estimation and masked reconstruction. The framework integrates a teacher-student segmentation network with a pseudo-label refiner to identify unreliable predictions and reconstruct them into cleaner supervision signals. We also provided theoretical analysis establishing the mathematical conditions under which refinement improves pseudo-label quality. With this design, our method achieved state-of-the-art results on nuScenes-lidarseg and SemanticKITTI across various label ratios.

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

# A APPENDIX

## A.1 THEORETICAL ANALYSIS ON TASK DIFFICULTY

We investigate Proposition 1, describing the relationship between two tasks: the segmentation task and the refinement task, which refines pseudo-labels generated by another segmentation model.

**Definition 1** (Segmentation and Refinement Tasks). *Let $X$ denote the input 3D LiDAR point data and $Y$ the segmentation labels. The segmentation task $Z$ can be expressed as:*

$$Z : X \to Y, \tag{12}$$

*which predicts segmentation labels from the input data. Likewise, let $T$ represent additional features including segmentation predictions predicted by another network. The refinement task $Z'$ is formulated as:*

$$Z' : (X, T) \to Y, \tag{13}$$

*which predicts segmentation labels from the input data and additional features.*

**Lemma 1** (Conditional Entropy Quantifying Task Difficulty). *Assume the hypothesis spaces $\mathcal{H}_Z$ and $\mathcal{H}_{Z'}$ have comparable complexity with similar VC-dimensions (Vapnik, 1998). Under this assumption, the difficulty $D(\cdot)$ of a supervised task can be quantified using conditional entropy (Cover & Thomas, 2006). We obtain the difficulty of the two tasks from Definition 1 as:*

$$D(Z) = H(Y \mid X), \quad D(Z') = H(Y \mid X, T). \tag{14}$$

The proof of Proposition 1 is directly induced from Lemma 1.

*Proof.* By the chain rule of conditional entropy:

$$D(Z) - D(Z') = H(Y \mid X) - H(Y \mid X, T) \tag{15}$$

$$= H(Y \mid X) - (H(Y \mid X) - I(Y; T \mid X)) \tag{16}$$

$$= I(Y; T \mid X). \tag{17}$$

Since the mutual information $I(Y; T \mid X) \geq 0$ by definition, we obtain $D(Z') \leq D(Z)$ from Proposition 1, with equality if and only if $T$ provides no information about $Y$ beyond what is already contained in $X$ (Cover & Thomas, 2006). □

**Implication.** In a semi-supervised setting, however, $T$ conveys semantic cues such as tentative class assignments or boundary structures that are not directly available from $X$. Such signals make $T$ a valuable source of information. Empirical evidence confirms that incorporating pseudo-labels as additional inputs or supervision improves performance across various semi-supervised settings (Yao et al., 2020; Wang & Yao, 2021; Chen et al., 2024). Thus, the additional features $T$ can reduce the uncertainty in predicting $Y$, potentially lowering the difficulty of the refinement task compared to the original segmentation task. Under the conditions where pseudo-labels provide meaningful semantic information, the refinement task would be less challenging than the original segmentation task.

## A.2 THEORETICAL ANALYSIS ON IMPROVEMENT CONDITION

We investigate Proposition 2, which characterizes the condition under which refinement improves the quality of pseudo-labels.

**Definition 2** (Voxel Partitions and Metrics). *For the $j$-th scene with voxel grid $\Omega = \{1, \ldots, H\} \times \{1, \ldots, W\} \times \{1, \ldots, L\}$ of size $N = |\Omega|$, let $Q_j(\omega)$, $\hat{Q}_j(\omega)$, and $[Y_j]_\omega$ denote the teacher prediction, refiner prediction, and ground truth at voxel $\omega \in \Omega$, respectively. An error-candidate mask $M_j \in \{0, 1\}^{H \times W \times L}$ partitions $\Omega$ into*

$$E_j = \{\omega : [M_j]_\omega = 1\}, \qquad C_j = \{\omega : [M_j]_\omega = 0\}. \tag{18}$$

*We further split $E_j$ into misclassified voxels by the teacher:*

$$E_{j,\text{err}} = \{\omega \in E_j : Q_j(\omega) \neq [Y_j]_\omega\}, \tag{19}$$

*and correctly classified voxels still marked as unreliable:*

$$E_{j,\text{cor}} = \{\omega \in E_j : Q_j(\omega) = [Y_j]_\omega\} = E_j \setminus E_{j,\text{err}}. \tag{20}$$

*On these partitions, we introduce the following quantities:*

$$\pi_j = \frac{|E_{j,\text{err}}|}{|E_j|}, \qquad \rho_j = \frac{|E_j|}{N}, \tag{21}$$

*where $\pi_j$ is the fraction of unreliable voxels that the teacher misclassifies, while $\rho_j$ is the relative size of the unreliable region compared to the entire scene.*

*We also define the correction and the error introduction rates, which measure the refiner's performance on the unreliable region $E_j$. The correction rate $q_j$ represents the fraction of voxels in $E_{j,\text{err}}$ that the refiner successfully corrects to match the ground truth. Conversely, the error introduction rate $r_j$ is the fraction of voxels in $E_{j,\text{cor}}$ that the refiner mistakenly changes away from the ground truth:*

$$q_j = \frac{|\{\omega \in E_{j,\text{err}} : \hat{Q}_j(\omega) = [Y_j]_\omega\}|}{|E_{j,\text{err}}|}, \quad r_j = \frac{|\{\omega \in E_{j,\text{cor}} : \hat{Q}_j(\omega) \neq [Y_j]_\omega\}|}{|E_{j,\text{cor}}|}. \tag{22}$$

**Definition 3** (Accuracy of Predictions). *The baseline accuracy of the teacher on the $j$-th scene is defined as the fraction of correctly predicted voxels:*

$$\text{Acc}_{\text{base}}(j) = \frac{1}{N} \sum_{\omega \in \Omega} \mathbf{1}_{\{Q_j(\omega)=[Y_j]_\omega\}}. \tag{23}$$

*With the refinement, predictions for voxels in $C_j$ remain unchanged, while those in $E_j$ are replaced with the refiner's outputs. The refined accuracy is:*

$$\text{Acc}_{\text{repl}}(j) = \frac{1}{N} \left( \sum_{\omega \in C_j} \mathbf{1}_{\{Q_j(\omega)=[Y_j]_\omega\}} + \sum_{\omega \in E_j} \mathbf{1}_{\{\hat{Q}_j(\omega)=[Y_j]_\omega\}} \right). \tag{24}$$

We now show Proposition 2 follows from the difference between these two accuracies by calculating on each partition in Definition 2.

*Proof.* We compute the relative accuracy improvement by subtracting the baseline from the refined accuracy:

$$\Delta_j = \text{Acc}_{\text{repl}}(j) - \text{Acc}_{\text{base}}(j) \tag{25}$$

$$= \frac{1}{N} \left( \sum_{\omega \in C_j} \mathbf{1}_{\{Q_j(\omega)=[Y_j]_\omega\}} + \sum_{\omega \in E_j} \mathbf{1}_{\{\hat{Q}_j(\omega)=[Y_j]_\omega\}} \right) - \frac{1}{N} \sum_{\omega \in \Omega} \mathbf{1}_{\{Q_j(\omega)=[Y_j]_\omega\}}. \tag{26}$$

Since $C_j$ and $E_j$ form a partition of $\Omega$, we have:

$$\frac{1}{N} \sum_{\omega \in \Omega} \mathbf{1}_{\{Q_j(\omega)=[Y_j]_\omega\}} = \frac{1}{N} \left( \sum_{\omega \in C_j} \mathbf{1}_{\{Q_j(\omega)=[Y_j]_\omega\}} + \sum_{\omega \in E_j} \mathbf{1}_{\{Q_j(\omega)=[Y_j]_\omega\}} \right). \tag{27}$$

Thus, the improvement simplifies to:

$$\Delta_j = \frac{1}{N} \sum_{\omega \in E_j} \left\{ \mathbf{1}_{\{\hat{Q}_j(\omega)=[Y_j]_\omega\}} - \mathbf{1}_{\{Q_j(\omega)=[Y_j]_\omega\}} \right\}. \tag{28}$$

Further partitioning $E_j$ into $E_{j,\text{err}}$ and $E_{j,\text{cor}}$:

$$\Delta_j = \frac{1}{N} \sum_{\omega \in E_{j,\text{err}}} \left\{ \mathbf{1}_{\{\hat{Q}_j(\omega)=[Y_j]_\omega\}} - \mathbf{1}_{\{Q_j(\omega)=[Y_j]_\omega\}} \right\} \tag{29}$$

$$+ \frac{1}{N} \sum_{\omega \in E_{j,\text{cor}}} \left\{ \mathbf{1}_{\{\hat{Q}_j(\omega)=[Y_j]_\omega\}} - \mathbf{1}_{\{Q_j(\omega)=[Y_j]_\omega\}} \right\}. \tag{30}$$

By Definition 2, for $\omega \in E_{j,\mathrm{err}}$: $\mathbf{1}_{\{Q_j(\omega)=[Y_j]_\omega\}} = 0$, and for $\omega \in E_{j,\mathrm{cor}}$: $\mathbf{1}_{\{Q_j(\omega)=[Y_j]_\omega\}} = 1$. This yields:

$$\Delta_j = \frac{1}{N} \sum_{\omega \in E_{j,\mathrm{err}}} \mathbf{1}_{\{\hat{Q}_j(\omega)=[Y_j]_\omega\}} + \frac{1}{N} \sum_{\omega \in E_{j,\mathrm{cor}}} \left\{ \mathbf{1}_{\{\hat{Q}_j(\omega)=[Y_j]_\omega\}} - 1 \right\} \tag{31}$$

$$= \frac{1}{N} \cdot |\{\omega \in E_{j,\mathrm{err}} : \hat{Q}_j(\omega) = [Y_j]_\omega\}| - \frac{1}{N} \cdot |\{\omega \in E_{j,\mathrm{cor}} : \hat{Q}_j(\omega) \neq [Y_j]_\omega\}| \tag{32}$$

$$= \rho_j \cdot \pi_j \cdot q_j - \rho_j \cdot (1 - \pi_j) \cdot r_j \tag{33}$$

$$= \rho_j \left( \pi_j q_j - (1 - \pi_j) r_j \right). \tag{34}$$

For accuracy improvement, we require $\Delta_j > 0$. Since $\rho_j > 0$, this is equivalent to:

$$\pi_j q_j - (1 - \pi_j) r_j > 0 \quad \Rightarrow \quad \pi_j (q_j + r_j) - r_j > 0. \tag{35}$$

When $q_j + r_j > 0$, we obtain:

$$\zeta_j := \pi_j - \frac{r_j}{q_j + r_j} > 0, \tag{36}$$

where $\zeta_j$ defines the improvement condition. $\qquad\square$

### A.3 Experiments on Improvement Conditions

We empirically validated the improvement condition $\zeta$ in Eq. (36), using values averaged over the validation sets of nuScenes-lidarseg (Fong et al., 2021) and SemanticKITTI (Behley et al., 2019), as reported in Table 8. As shown in the table, $\zeta$ remained strictly positive across all experimental settings, confirming that refinement consistently operates in a regime where accuracy improvements are guaranteed. Notably, the results further suggest that even a simple error estimation strategy can satisfy the condition, enabling REPL to reliably improve pseudo-label quality.

Table 8: Improvement conditions on nuScenes-lidarseg and SemanticKITTI under varying the ratio of labeled data (1%, 10%, 20%, 50%).

| Method | nuScenes-lidarseg (Fong et al., 2021) | | | | SemanticKITTI (Behley et al., 2019) | | | |
|---|---|---|---|---|---|---|---|---|
| | 1% | 10% | 20% | 50% | 1% | 10% | 20% | 50% |
| *Improvement condition* $\zeta$ | 0.43 | 0.39 | 0.41 | 0.40 | 0.65 | 0.75 | 0.78 | 0.67 |

### A.4 A Comparison of Pseudo-Label Quality Between Teacher and Teacher With Refiner Over Time.

We additionally report the trends in pseudo-label quality for the teacher model and the teacher-with-refiner model during training on each unlabeled set of nuScenes-lidarseg in Figure 6. Across all label ratios in nuScenes-lidarseg (1%, 10%, 20%, 50%), we observe a consistent pattern; the refiner provides additional improvements over the EMA teacher throughout training. This suggests that the gains are not solely due to EMA updates but also reflect the contribution of the refinement process.

### A.5 Additional Qualitative Results

We present additional qualitative results on the unlabeled data of nuScenes-lidarseg (Fong et al., 2021) and SemanticKITTI (Behley et al., 2019) with the ratio of labeled data 1%. We show comparisons between refined pseudo-labels and their initial versions in Figure 8 and Figure 9. The results demonstrate that the refinement process effectively reduces noise and corrects errors in the initial pseudo-labels. We also visualize results on the validation set of each dataset in Figure 10 and Figure 11. Additionally, Figure 12 and Figure 13 provide detailed comparisons focusing on long-range regions, where the refinement process shows particularly significant improvements in handling challenging scenarios with reduced noise and error correction.

### A.6 Disclosure of the Use of Large Language Models

We used Large Language Models (LLMs) solely to aid and polish the writing of this paper. LLMs did not contribute to research ideation, experimental design, or analysis. The authors take full responsibility for all content.

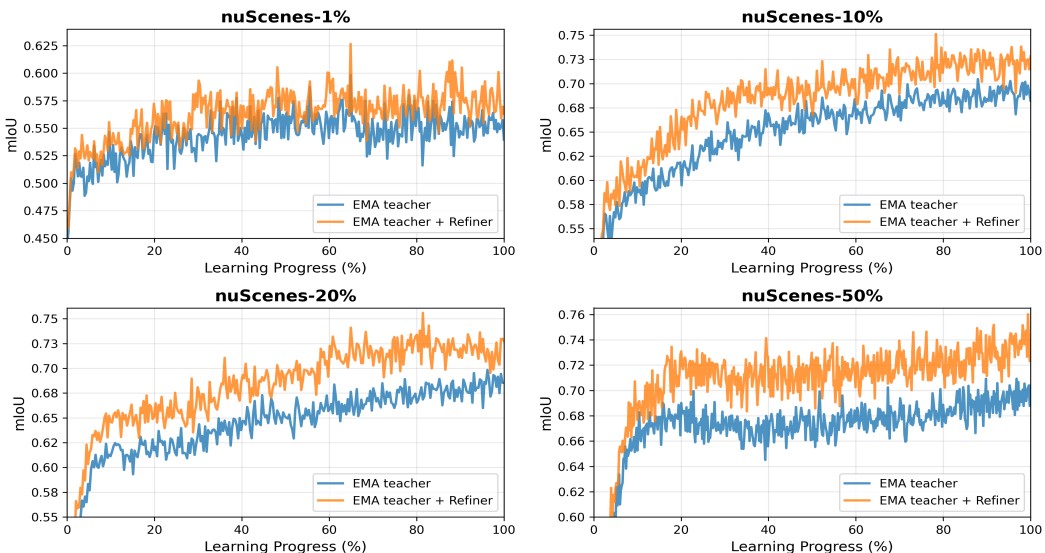

Figure 6: Comparison of pseudo-label quality between the teacher model and the teacher-with-refiner model during training on each unlabeled set of nuScenes-lidarseg.

## A.7 ADDITIONAL ANALYSIS ON PSEUDO-LABEL QUALITY IMPROVEMENT THROUGHOUT TRAINING.

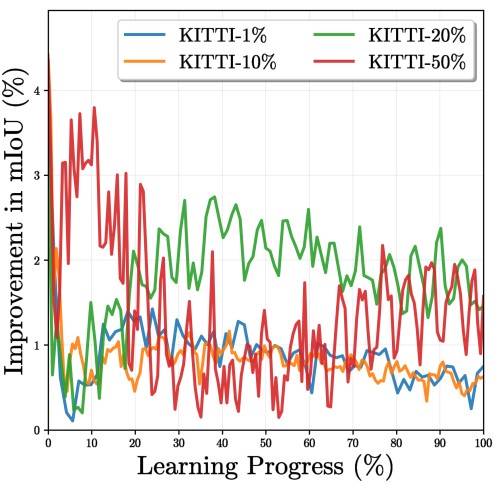

Figure 7: Pseudo-label quality improvement by the refiner during training on SemanticKITTI.

We additionally illustrate the pseudo-label quality improvement throughout training on SemanticKITTI across different labeled data ratios (1%, 10%, 20%, 50%) in Figure 7. Initially, improvements were modest across almost all settings since the refiner has limited knowledge for effective error correction. As training progresses, the refiner developed stronger error correction capabilities, leading to more substantial improvements. However, these gains gradually diminished in later training stages as the base model becomes increasingly accurate, leaving fewer errors to correct. REPL consistently delivered benefits across all label ratios, though the improvements on SemanticKITTI show greater variability compared to nuScenes-lidarseg, likely due to inherent dataset characteristics and complexity differences.

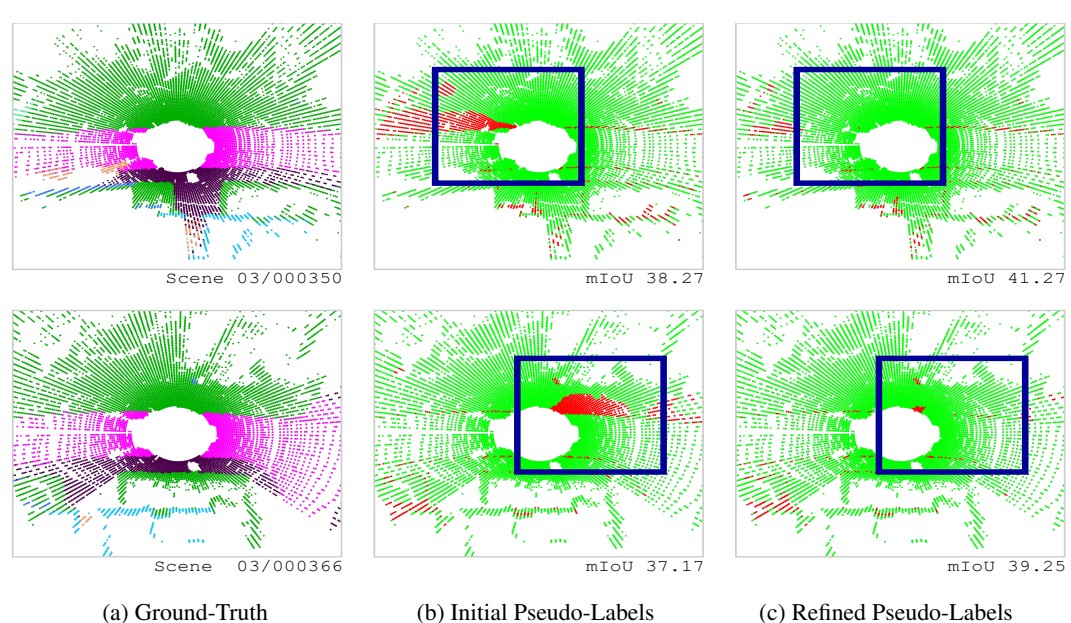

|                    |                      |                     |
| :----------------: | :------------------: | :-----------------: |
| (a) Ground-Truth   | (b) Initial Pseudo-Labels | (c) Refined Pseudo-Labels |

Figure 8: Qualitative results of refined pseudo-labels and their initial predictions on the unlabeled data of SemanticKITTI. Correct and incorrect predictions are shown in green and red, respectively. The model was trained with a 1% label ratio.

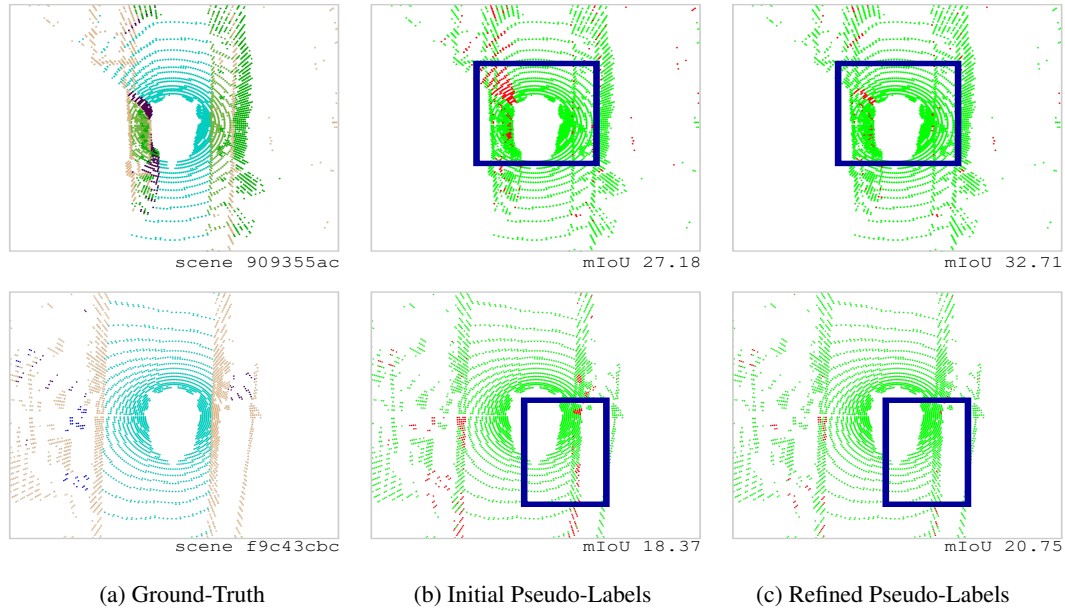

|                    |                      |                     |
| :----------------: | :------------------: | :-----------------: |
| (a) Ground-Truth   | (b) Initial Pseudo-Labels | (c) Refined Pseudo-Labels |

Figure 9: Qualitative results of refined pseudo-labels and their initial predictions on the unlabeled data of nuScenes-lidarseg. Correct and incorrect predictions are shown in green and red, respectively. The model was trained with a 1% label ratio.

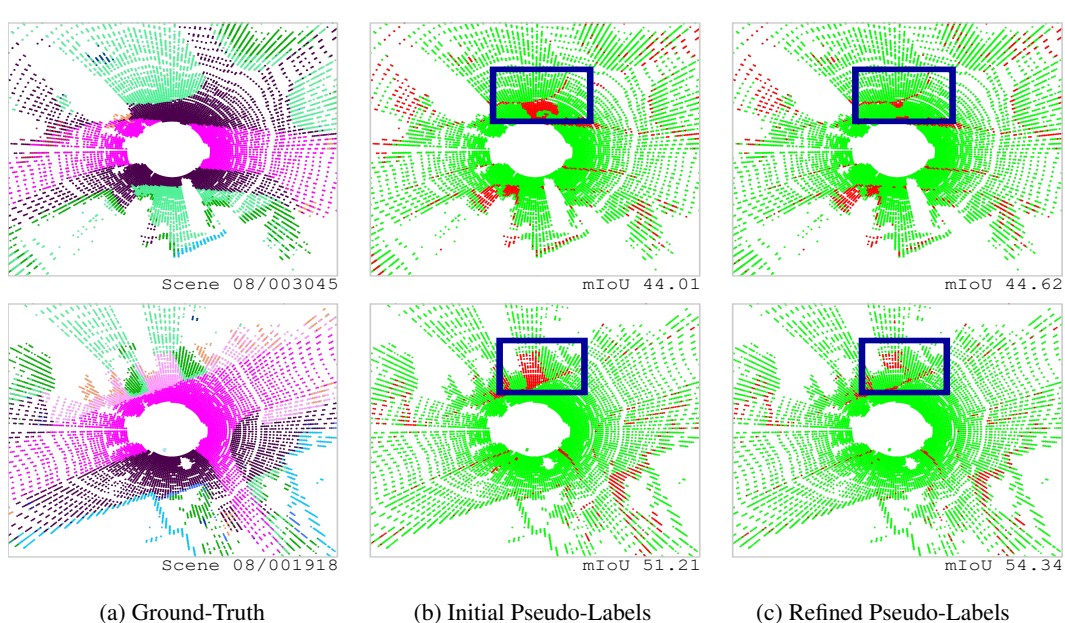

(a) Ground-Truth  (b) Initial Pseudo-Labels  (c) Refined Pseudo-Labels

Figure 10: Qualitative results of refined pseudo-labels and their initial predictions on the validation set of SemanticKITTI. Correct and incorrect predictions are shown in green and red, respectively. The model was trained with a 1% label ratio.

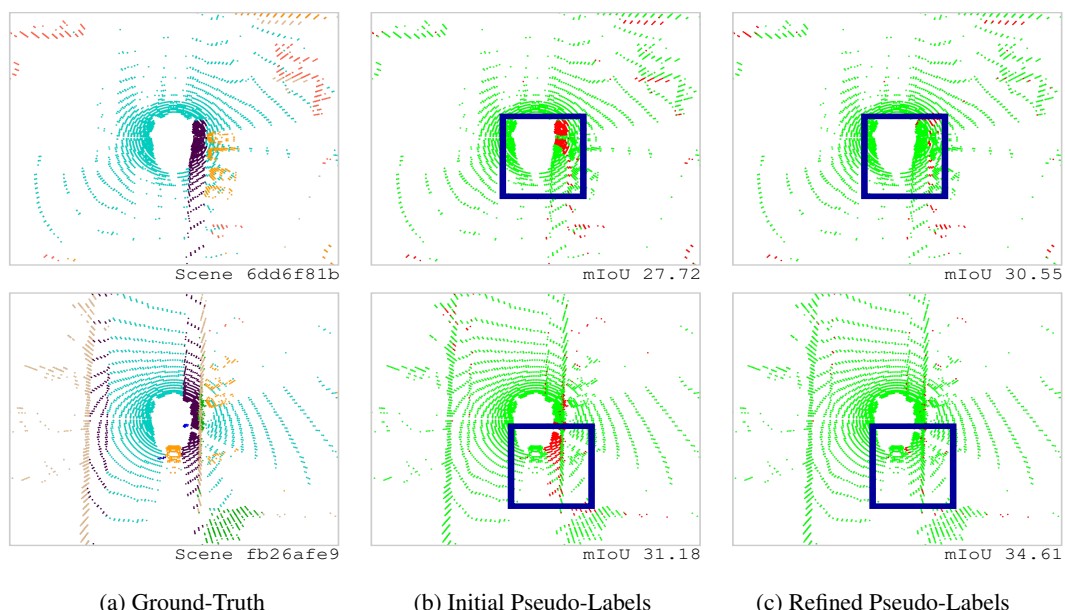

(a) Ground-Truth  (b) Initial Pseudo-Labels  (c) Refined Pseudo-Labels

Figure 11: Qualitative results of refined pseudo-labels and their initial predictions on the validation set of nuScenes-lidarseg. Correct and incorrect predictions are shown in green and red, respectively. The model was trained with a 1% label ratio.

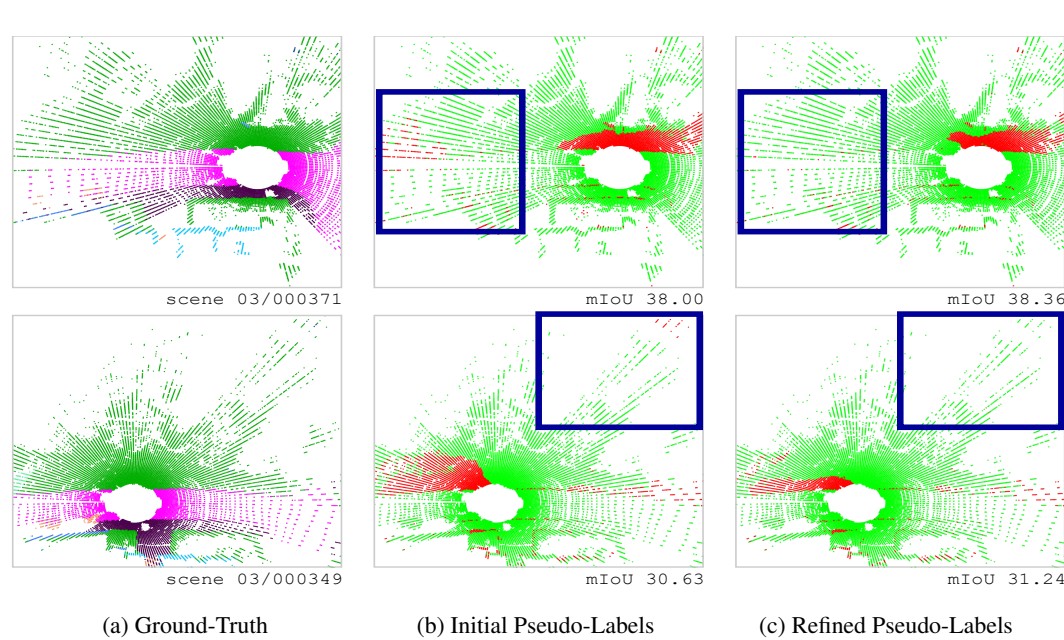

|     (a) Ground-Truth     |     (b) Initial Pseudo-Labels     |     (c) Refined Pseudo-Labels     |

Figure 12: Qualitative results of refined pseudo-labels and their initial predictions on the unlabeled set of SemanticKITTI. Correct and incorrect predictions are shown in green and red, respectively. The model was trained with a 1% label ratio.

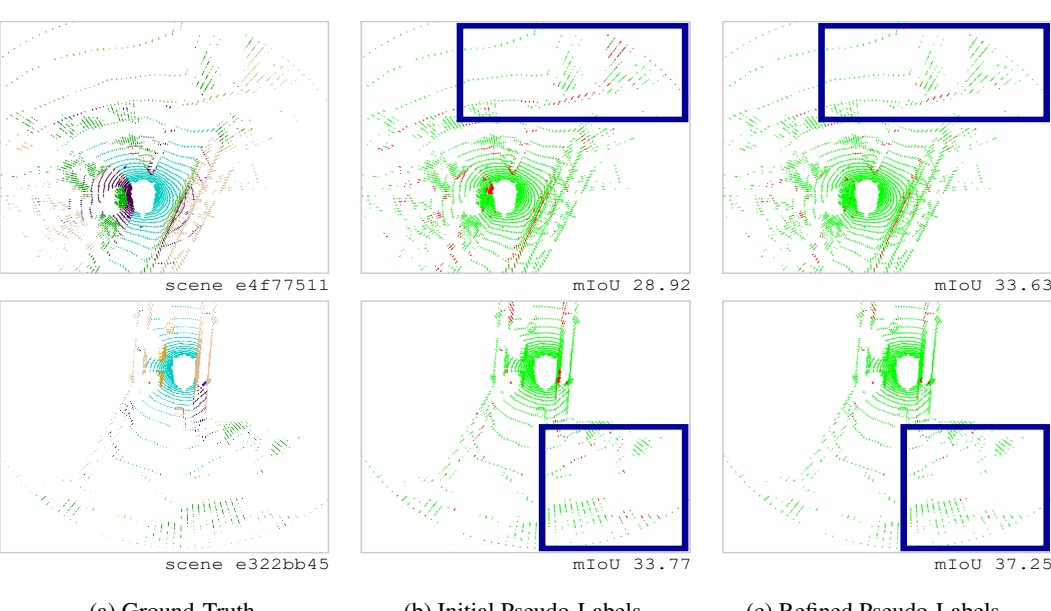

|     (a) Ground-Truth     |     (b) Initial Pseudo-Labels     |     (c) Refined Pseudo-Labels     |

Figure 13: Qualitative results of refined pseudo-labels and their initial predictions on the unlabeled set of nuScenes-lidarseg. Correct and incorrect predictions are shown in green and red, respectively. The model was trained with a 1% label ratio.

