# OpenReview forum: "RePL: Pseudo-label Refinement for Semi-supervised LiDAR Semantic Segmentation"
_ICLR.cc/2026/Conference — Submitted to ICLR 2026_

### Official Review · Reviewer_X4n8 · 2025-10-25

**Soundness:** 3
**Presentation:** 3
**Contribution:** 3
**Rating:** 8
**Confidence:** 4

**Summary:**

This manuscript addresses a long-standing issue in semi-supervised LiDAR semantic segmentation: the error propagation and confirmation bias from noisy pseudo-labels. The authors propose REPL, a pseudo-label refinement framework that integrates a teacher-student model with a novel refiner module.

The refiner detects unreliable pseudo-labels using confidence-based teacher–student agreement, then corrects them via masked reconstruction inspired by masked autoencoders. The framework is complemented by theoretical analysis showing conditions under which refinement improves pseudo-label quality, as well as a training strategy with random masking, negative learning, and mixed labeled/unlabeled scenes to strengthen supervision.

Empirical results on nuScenes-lidarseg and SemanticKITTI benchmarks demonstrate consistent improvements over state-of-the-art semi-supervised methods across various label ratios.

**Strengths:**

(+) Tackling noisy pseudo-labels is highly relevant, as confirmation bias remains a fundamental bottleneck in semi-supervised LiDAR semantic segmentation. This manuscript articulates this motivation clearly and positions REPL against existing post-hoc strategies. The idea of refining pseudo-labels before training (rather than adjusting them afterward via weighting or filtering) provides a clean, conceptually different direction. The masked reconstruction approach is well-aligned with recent advances in self-supervised learning.

(+) Theoretical justification: The inclusion of theoretical results (Propositions 1 and 2) formalizes why refinement is easier than generating labels from scratch and under what conditions it yields improvements. This adds depth beyond empirical validation.

(+) REPL achieves SOTA on both nuScenes-lidarseg and SemanticKITTI, with notable gains especially at low label ratios (e.g., +2.0 mIoU over IT2 on average). The ablation studies and sensitivity analyses further support the design choices.

**Weaknesses:**

(-) While the refinement-based perspective is novel, many components (teacher-student EMA, confidence thresholds, masked reconstruction) are individually adapted from prior works. The contribution lies more in integration than in fundamentally new architectures.

(-) The reliability detection relies on simple agreement-based thresholds. Although results show effectiveness, this component may be brittle in challenging cases (e.g., rare classes, long-range sparse points). More adaptive or learned error-detection strategies could strengthen the approach.

(-) Evaluation comprehensiveness: Please consider comparing FRNet [Xu, et al.], LarseMix++ [Kong, et al.], and Lim3D [Li, et al.] for a more comprehensive evaluation on the semi-supervised LiDAR segmentation benchmark. Additionally, there are several self-supervised LiDAR segmentation methods that are evaluated on the same benchmark; please consider including those as well, such as SLidR [Sautier, et al.], Seal [Liu, et al.], and SuperFlow [Xu, et al.].

References:
- FRNet [Xu, et al.] FRNet: Frustum-Range Networks for Scalable LiDAR Segmentation. TIP, 2025.
- LarseMix++ [Kong, et al.] Multi-Modal Data-Efficient 3D Scene Understanding for Autonomous Driving. TPAMI, 2025.
- Lim3D [Li, et al.] Less is More: Reducing Task and Model Complexity for 3D Point Cloud Semantic Segmentation. CVPR, 2023.
- SLidR [Sautier, et al.] Image-to-Lidar Self-Supervised Distillation for Autonomous Driving Data. CVPR, 2022.
- Seal [Liu, et al.] Segment Any Point Cloud Sequences by Distilling Vision Foundation Models. NeurIPS, 2023.
- SuperFlow [Xu, et al.] 4D Contrastive Superflows are Dense 3D Representation Learners. ECCV, 2024.

---

**Minor Suggestions**

(-) Clarify computational overhead: The refiner introduces an additional network and losses—reporting training/inference cost overhead would be valuable for practitioners.

(-) Proposition 2 is well-explained, but its assumptions (e.g., independence of correction/error rates) could be elaborated for readers less familiar with information-theoretic analysis.

(-) While some qualitative comparisons are shown, additional side-by-side visualizations across challenging categories could better demonstrate where refinement helps most.

(-) Some sections could be slightly condensed for readability, especially in the theoretical analysis.

**Questions:**

The manuscript tackles a critical challenge in semi-supervised LiDAR semantic segmentation and offers a conceptually fresh perspective—refining pseudo-labels instead of merely reweighting or filtering them. The integration with masked reconstruction and theoretical grounding strengthens its contributions, and empirical results are solid.

That said, the core novelty is more in formulation and integration rather than new architectural design, and the evaluation scope is somewhat narrow. Nevertheless, the work is well-motivated, cleanly executed, and empirically convincing, making it a meaningful step forward for the community. With additional efforts in addressing the main weaknesses and minor suggestions (as detailed above), the manuscript could have a strong impact.

---

> ### Author Response · Authors · 2025-11-20
> **Official Comment by Authors (Part 1/4)**
>
> Thank you for the insightful and constructive comment! Below, we provide detailed responses to address your concerns.
>
> &nbsp;
>
> ### **Contribution of RePL**
>
> We appreciate your thoughtful comment.
>
> The individual components we use are indeed established techniques, and we do not claim them as standalone contributions. Our contribution is in introducing, to the best of our knowledge, the first refinement framework specifically designed for 3D LiDAR pseudo labels, demonstrating that masked reconstruction is a powerful tool in this context. We also would claim that our theoretical analyses are also a notable contribution, offering a strong impact on label-efficient LiDAR segmentation.
>
> We below discuss our contributions in more detail.
>
> - We clarify that the agreement-based unreliable voxel identification is made so simple on purpose to better demonstrate the power of our refinement method, and not claimed as a core contribution. It functions as a minimal mechanism to expose the refinement process to informative signals, and any advanced unreliable voxel detection methods can further improve the performance as demonstrated in Table 4 of the main paper.
>
> - One of the main contributions lies in using masked reconstruction as a refiner for correcting 3D LiDAR pseudo labels. Masked reconstruction is commonly used as a representation learning objective, but applying it as a label correction operator in sparse voxelized LiDAR requires dedicated designs specific to this domain. We believe that bringing this mechanism into LiDAR segmentation and demonstrating its outstanding performance is clearly a contribution of our work.
>
> - To our knowledge, our work is the first to formulate and systematize pseudo label refinement for 3D LiDAR segmentation. Prior methods rely on consistency between teacher and student without an explicit refinement module. Our formulation defines unreliable region identification, reconstruction-based correction, and the separation of roles among components, providing a clear framework for pseudo-label refinement in LiDAR segmentation.
>
> - We provide theoretical analyses explaining why reconstruction-driven refinement can correct label noise under realistic conditions. These analyses support our idea theoretically and distinguish our work from a simple aggregation of existing ideas.
>
> &nbsp;

---

> ### Author Response · Authors · 2025-11-20
> **Official Comment by Authors (Part 2/4)**
>
> ### **Robustness to Class Imbalance and Future Potential**
>
> Thank you for the insightful comment.
>
> We clarify that the confidence-based unreliable voxel identification is not a core contribution of our work; it is intentionally kept simple to demonstrate the effectiveness of our refinement mechanism itself. Despite its simplicity, this agreement-based strategy is already sufficient to reach strong performance across diverse scenarios, including challenging cases such as rare classes and long-range sparse points.
>
> Our experiments show that the refinement framework achieves notable performance gains even for rare and challenging classes. The tables below present per-class improvements of pseudo-labels on unlabeled data from SemanticKITTI and nuScenes-lidarseg trained with 1% labeled ratio, along with the relative frequency of each class:
>
> | Class ID | Class Name | Performance of Initial Pseudo-Labels (IoU) | Performance of refined Pseudo-Labels (IoU) | Frequency (%) | Difference |
> | ------------------------------------------------------------ | ------------- | --------- | ----------- | ------------- | ---------- |
> | 1 | car | 91.99 | 92.94 |  4.399  | 0.95 |
> | 2 | bicycle | 32.22 | 34.08 |  0.0166  | 1.86 |
> | 3 | motorcycle | 25.81 | 28.24 |  0.0411  | 2.43 |
> | 4 | truck | 51.43 | 53.81 |  0.2236  | 2.38 |
> | 5 | other-vehicle | 31.99 | 33.72 |  0.1866  | 1.73 |
> | 6 | person | 55.90 | 59.63 |  0.0349  | 3.73 |
> | 7 | bicyclist | 48.73 | 50.18 |  0.0131  | 1.45 |
> | 8 | motorcyclist | 59.02 | 58.08 |  0.00387  | -0.94 |
> | 9 | road | 93.57 | 93.98 |  20.526  | 0.41 |
> | 10 | parking | 66.48 | 68.51 |  1.520  | 2.03 |
> | 11 | sidewalk | 82.90 | 84.05 |  14.860  | 1.15 |
> | 12 | other-ground | 51.17 | 54.87 |  0.403  | 3.70 |
> | 13 | building | 90.65 | 91.55 |  13.700  | 0.90 |
> | 14 | fence | 66.75 | 68.09 |  7.471  | 1.34 |
> | 15 | vegetation | 84.22 | 85.44 |  27.549  | 1.22 |
> | 16 | trunk | 71.42 | 74.15 |  0.623  | 2.73 |
> | 17 | terrain | 71.07 | 72.73 |  8.068  | 1.66 |
> | 18 | pole | 63.46 | 66.31 |  0.295  | 2.85 |
> | 19 | traffic-sign | 63.30 | 64.09 |  0.0636  | 0.79 |
> | **Average** | | **63.27** | **64.97** | – | **1.70** |
>
> **Table R1.** Per-class improvements of pseudo-labels on unlabeled data from SemanticKITTI with 1% labeled ratio.
>
> | Class ID | Class Name | Performance of Initial Pseudo-Labels (IoU) | Performance of refined Pseudo-Labels (IoU) | Frequency (%) | Difference |
> | ------------------------------------------------------------ | -------------------- | --------- | ----------- | ------------- | ---------- |
> | 1 | barrier | 60.22 | 63.67 |  1.10  | 3.45 |
> | 2 | bicycle | 10.74 | 13.42 |  0.02  | 2.68 |
> | 3 | bus | 60.10 | 61.14 |  0.55  | 1.04 |
> | 4 | car | 85.82 | 87.42 |  4.42  | 1.60 |
> | 5 | construction_vehicle | 14.89 | 17.43 |  0.18  | 2.54 |
> | 6 | motorcycle | 38.61 | 41.16 |  0.05  | 2.55 |
> | 7 | pedestrian | 59.97 | 66.44 |  0.27  | 6.47 |
> | 8 | traffic_cone | 51.29 | 59.15 |  0.09  | 7.86 |
> | 9 | trailer | 53.88 | 54.48 |  0.58  | 0.60 |
> | 10 | truck | 62.08 | 63.10 |  1.83  | 1.02 |
> | 11 | driveable_surface | 94.82 | 93.35 |  37.58  | -1.47 |
> | 12 | flat_other | 51.77 | 54.37 |  1.02  | 2.60 |
> | 13 | sidewalk | 69.36 | 66.96 |  8.31  | -2.40 |
> | 14 | terrain | 67.53 | 71.67 |  8.34  | 4.14 |
> | 15 | manmade | 85.33 | 89.74 |  21.14  | 4.41 |
> | 16 | vegetation | 84.24 | 89.59 |  14.54  | 5.35 |
> | **Average** | | **59.42** | **62.07** | – | **2.65** |
>
> **Table R2.** Per-class improvements of pseudo-labels on unlabeled data from nuScenes-lidarseg with 1% labeled ratio.
>
> As demonstrated, several underrepresented and challenging categories (e.g., pedestrian, traffic_cone, person, bicycle) show particularly large gains, highlighting the robustness of our refinement approach even with simple agreement-based detection. Furthermore, we included in-depth visual comparisons in Figure 12 and Figure 13 that examine long-range scenarios, where our refinement approach demonstrates notable effectiveness in mitigating noise and correcting segmentation errors. Extended discussion of these results can be found in the appendix (line 856).
>
> We agree that more adaptive or learned error-detection strategies have the potential to further improve performance. As shown in Table 4 of the main paper (line 466), there is substantial room for additional gains if the error detection signal becomes more accurate, indicating that our current choice is not close to the upper bound. More advanced detection methods tailored to handle class imbalance or distribution shifts can be readily integrated into our framework for further improvements. We view exploring such detectors as a promising direction and a natural extension of the framework rather than a limitation of the current design.

---

> ### Author Response · Authors · 2025-11-20
> **Official Comment by Authors (Part 3/4)**
>
> ### **Broader Comparison with Additional Methods**
>
> Thank you for this helpful suggestion.
>
> We agree that a broader comparison can provide a more complete view of the semi-supervised LiDAR segmentation landscape. In our main experiments, we focused on methods that are directly comparable under the same backbone and training protocol, since fair evaluation is difficult when architectural assumptions or input modalities differ. Under this criterion, we included methods such as LaserMix++ and FRNet, which use Cylinder3D and align well with our setting.
>
> For the remaining methods you mentioned, we also conducted a careful review and included them in a separate comparison table with clear notes on the differences in backbone, supervision signals, and auxiliary modalities.
> Although these methods are not directly comparable, we appreciate the suggestion and have reported their numbers with the appropriate caveats to improve completeness in Table 1 of the revised paper (line 368).
>
> &nbsp;
>
> ### **Computational Cost & Efficiency**
>
> To quantify the additional overhead by the refiner, we measured the latency and memory usage during inference on the validation set using a single batch.
>
> As shown in the table below, the refiner adds approximately 0.25 seconds of latency and 396 MB of memory, while providing a substantial improvement of +9.1 mIoU from the supervised-only baseline. These results demonstrate that the added computational cost is moderate relative to the significant accuracy gains.
>
> | **Method**                  | **Latency(s)** | **Memory (MB)** | **mIoU**  |
> | --------------------------- | -------------- | --------------- | --------- |
> | Baseline (supervised-only)  | 0.43           | 1231            | 50.9      |
> | Baseline+Refiner (semi-sup) | 0.68           | 1627            | 60.0      |
> | Δ                           | +0.25s         | +396 MB         | +9.1 mIoU |
>
> **Table R3.** Computational cost analysis on nuScenes-lidarseg.
>
> We have described this results in the revision (Table 7, line 478)

---

> ### Author Response · Authors · 2025-11-20
> **Official Comment by Authors (Part 4/4)**
>
> ### **Elaboration of Proposition 2 and Theoretical Analysis Condensation for Readability**
>
> Thank you for the constructive comment.
>
> We have revised the explanation of Proposition 2 in the main text to improve clarity and readability (line 301, line 765). Specifically, we have added detailed descriptions of the correction rate $q_j$ and error rate $r_j$, and provided explicit connections to how the refiner operates on unreliable pseudo-labels in our framework.
>
> Regarding the assumptions, we clarify that while Proposition 2 is stated in terms of correction and error rates, these parameters can be empirically estimated from the training data according to their definitions. For instance, $q_j$ can be measured as the fraction of originally incorrect pseudo-labels that are corrected by the refiner, and $r_j$ as the fraction of originally correct labels that are corrupted. Since these quantities are observable from data, the proposition does not require strong independence assumptions or restrictive theoretical constraints, and one can simply substitute the measured values and verify the condition. This makes the proposition practically applicable with minimal assumptions.
>
> We believe these clarifications address the concern about accessibility for readers less familiar with information-theoretic analysis, while maintaining the theoretical soundness of the result. If we have misunderstood your concern or if further clarification would be helpful, please let us know and we will be happy to provide additional revisions.
>
>
> &nbsp;
>
> ### **Additional Visualizations Across Challenging Categories**
>
> Thank you for this suggestion.
>
> Visual comparisons focusing on difficult cases would indeed provide clearer insights into the effectiveness of our refinement approach. To address this, we included additional qualitative results in Figure 4 of the revised paper that focus explicitly on failure cases, with a separate paragraph discussing these results (line 527).
>
> Furthermore, we present in-depth visual comparisons in Figure 12 and Figure 13 that examine challenging scenarios, including long-range regions. These visualizations demonstrate where our refinement approach is notably effective in mitigating noise and correcting segmentation errors. Extended discussion of these results can be found in the appendix (line 856).
>
> We also provide quantitative analyses in **Table R1** and **R2** for both SemanticKITTI and nuScenes-lidarseg, which show class-wise performance changes. The improvements are consistent for most object classes, with particularly notable gains in challenging categories such as pedestrians, traffic cones, and bicycles despite their limited point density. These additions provide a more complete picture of where REPL demonstrates its strengths and clarify its behavior across different category types.

---

> ### Author Response · Authors · 2025-11-26
> **Gentle Reminder**
>
> Thank you once again for your thoughtful review. Your feedback has been incredibly helpful in improving our work.
>
> As the discussion period is nearing its end, we would be grateful if you could review the modifications and additional analysis we have made in response to your concerns. We hope these revisions address the points you raised and would appreciate your feedback at your convenience.
>
> We sincerely appreciate your valuable time and effort in reviewing our work.

---

> ### Comment · Reviewer_X4n8 · 2025-11-27
>
> First of all, thanks to the authors for putting in the effort to address the raised concerns.
>
> I have read the authors' responses, as well as the other two reviewers' comments. Most of the concerns raised from my side have been well addressed. Specifically, the authors have conducted an extra experiment on additional datasets and verified the strong robustness of REPL. Some more ablation results support the designed components in the overall framework.
>
> The quality of this work has been improved, and, to my sense, resembles the standard of the ICLR community. It could benefit 3D scene understanding and related autonomous driving domains. Therefore, I am leaning to maintain the positive rating and recommend acceptance.

---

> > ### Author Response · Authors · 2025-11-27
> > **Thank you!**
> >
> > Thank you for your time and the positive feedback! We're glad to hear that your main concerns have been addressed. If you have any remaining questions, just let us know and we'll be glad to help.

---

### Official Review · Reviewer_yQAk · 2025-10-30

**Soundness:** 3
**Presentation:** 3
**Contribution:** 3
**Rating:** 6
**Confidence:** 5

**Summary:**

The paper proposes REPL, a framework for semi-supervised LiDAR semantic segmentation. The main idea is to directly refine pseudo-labels rather than simply discarding low-confidence ones or reweighting them.

A teacher-student setup produces pseudo-labels, and a refiner detects unreliable voxels (via teacher–student confidence agreement) and reconstructs them using a masked autoencoder-inspired method. The authors also provide a theoretical analysis, showing when refinement improves label quality.

Experiments on nuScenes and SemanticKITTI benchmarks demonstrate strong performance, setting a new state of the art across different label ratios.

**Strengths:**

- Instead of post-hoc filtering or reweighting, REPL improves pseudo-labels at their source, directly tackling confirmation bias. The entropy-based task difficulty and the ζ condition provide a principled reason why refinement is beneficial. This is relatively rare in semi-supervised segmentation papers.

- New SOTA on nuScenes and SemanticKITTI, with robust gains especially in the low-label regime (e.g., +4.7 mIoU over supervised-only baseline at 1% labeled data). Ablation studies show the importance of each loss (Lrsup, Lrunl, Lrmix), and sensitivity experiments (error mask quality, random masking) demonstrate the design choices matter.

- The paper is overall well-written. The figures (especially Fig. 1 and qualitative results in Fig. 3–9) make the method understandable, showing how refinement corrects pseudo-label noise.

**Weaknesses:**

- Teacher–student EMA, agreement-based confidence, and masked reconstruction are all known individually. The contribution is more in how they are combined and applied to LiDAR pseudo-label refinement.

- The confidence percentile rule is simple but might not be optimal in cases of class imbalance or rare categories (e.g., pedestrians, bicycles). Results could be brittle under distribution shifts.

- Evaluation is limited to nuScenes and SemanticKITTI. More diverse benchmarks (Waymo, SynLiDAR, Robo3D, or other datasets) would strengthen claims of generality.

- While qualitative improvements are shown, the paper doesn’t deeply discuss where REPL still fails—e.g., does it help small-object categories or long-range sparse regions?

**Questions:**

1. What is the additional training and inference cost of running the refiner? Is REPL practical for large-scale deployment?

2. How does REPL perform on rare classes or long-tail categories in SemanticKITTI (e.g., “motorcyclist,” “other-ground”)? Does the refinement help there or mostly on dominant classes?

3. Would a learned error detector (instead of heuristic agreement) further improve the error mask? Or is the current approach already close to optimal?

4. Can the refinement process be extended to sequential settings (e.g., 4D spatio-temporal LiDAR), where temporal consistency might aid error correction?

---

> ### Author Response · Authors · 2025-11-20
> **Official Comment by Authors (Part 1/5)**
>
> Thank you for the insightful and constructive comment! Below, we provide detailed responses to address your concerns.
>
> &nbsp;
>
> ### **Contribution of RePL**
>
> Thank you for the thoughtful comment.
>
> The individual components we use are indeed established techniques, and we do not claim them as standalone contributions. Our contribution is in introducing, to the best of our knowledge, the first refinement framework specifically designed for 3D LiDAR pseudo labels, demonstrating that masked reconstruction is a powerful tool in this context. We also would claim that our theoretical analyses are also a notable contribution, offering a strong impact on label-efficient LiDAR segmentation.
>
> We below discuss our contributions in more detail.
>
> - We clarify that the agreement-based unreliable voxel identification is made so simple on purpose to better demonstrate the power of our refinement method, and not claimed as a core contribution. It functions as a minimal mechanism to expose the refinement process to informative signals, and any advanced unreliable voxel detection methods can further improve the performance as demonstrated in Table 4 of the main paper.
>
> - One of the main contributions lies in using masked reconstruction as a refiner for correcting 3D LiDAR pseudo labels. Masked reconstruction is commonly used as a representation learning objective, but applying it as a label correction operator in sparse voxelized LiDAR requires dedicated designs specific to this domain. We believe that bringing this mechanism into LiDAR segmentation and demonstrating its outstanding performance is clearly a contribution of our work.
>
> - To our knowledge, our work is the first to formulate and systematize pseudo label refinement for 3D LiDAR segmentation. Prior methods rely on consistency between teacher and student without an explicit refinement module. Our formulation defines unreliable region identification, reconstruction-based correction, and the separation of roles among components, providing a clear framework for pseudo-label refinement in LiDAR segmentation.
>
> - We provide theoretical analyses explaining why reconstruction-driven refinement can correct label noise under realistic conditions. These analyses support our idea theoretically and distinguish our work from a simple aggregation of existing ideas.

---

> ### Author Response · Authors · 2025-11-20
> **Official Comment by Authors (Part 2/5)**
>
> ### **Robustness to Class Imbalance and Rare Categories**
>
> Thank you for the insightful comment.
>
> We clarify that the confidence-based unreliable voxel identification is not a core contribution of our work; it is intentionally kept simple to demonstrate the effectiveness of our refinement mechanism itself. As shown in Table 4, more advanced detection methods tailored to handle class imbalance or distribution shifts can be integrated into our framework for further improvements.
>
> Despite using this simple identification strategy, our experiments show that the refinement framework achieves notable performance gains even for rare classes. The tables below present per-class improvements of pseudo-labels on unlabeled data from SemanticKITTI and nuScenes-lidarseg trained with 1% labeled ratio with the relative frequency of each class ("frequency. (%)"):
>
> | Class ID | Class Name | Performance of Initial Pseudo-Labels (IoU) | Performance of refined Pseudo-Labels (IoU) | Frequency (%) | Difference |
> | ------------------------------------------------------------ | ------------- | --------- | ----------- | ------------- | ---------- |
> | 1 | car | 91.99 | 92.94 |  4.399  | 0.95 |
> | 2 | bicycle | 32.22 | 34.08 |  0.0166  | 1.86 |
> | 3 | motorcycle | 25.81 | 28.24 |  0.0411  | 2.43 |
> | 4 | truck | 51.43 | 53.81 |  0.2236  | 2.38 |
> | 5 | other-vehicle | 31.99 | 33.72 |  0.1866  | 1.73 |
> | 6 | person | 55.90 | 59.63 |  0.0349  | 3.73 |
> | 7 | bicyclist | 48.73 | 50.18 |  0.0131  | 1.45 |
> | 8 | motorcyclist | 59.02 | 58.08 |  0.00387  | -0.94 |
> | 9 | road | 93.57 | 93.98 |  20.526  | 0.41 |
> | 10 | parking | 66.48 | 68.51 |  1.520  | 2.03 |
> | 11 | sidewalk | 82.90 | 84.05 |  14.860  | 1.15 |
> | 12 | other-ground | 51.17 | 54.87 |  0.403  | 3.70 |
> | 13 | building | 90.65 | 91.55 |  13.700  | 0.90 |
> | 14 | fence | 66.75 | 68.09 |  7.471  | 1.34 |
> | 15 | vegetation | 84.22 | 85.44 |  27.549  | 1.22 |
> | 16 | trunk | 71.42 | 74.15 |  0.623  | 2.73 |
> | 17 | terrain | 71.07 | 72.73 |  8.068  | 1.66 |
> | 18 | pole | 63.46 | 66.31 |  0.295  | 2.85 |
> | 19 | traffic-sign | 63.30 | 64.09 |  0.0636  | 0.79 |
> | **Average** | | **63.27** | **64.97** | – | **1.70** |
>
> **Table R1.** Per-class improvements of pseudo-labels on unlabeled data from SemanticKITTI with 1% labeled ratio.
>
> | Class ID | Class Name | Performance of Initial Pseudo-Labels (IoU) | Performance of refined Pseudo-Labels (IoU) | Frequency (%) | Difference |
> | ------------------------------------------------------------ | -------------------- | --------- | ----------- | ------------- | ---------- |
> | 1 | barrier | 60.22 | 63.67 |  1.10  | 3.45 |
> | 2 | bicycle | 10.74 | 13.42 |  0.02  | 2.68 |
> | 3 | bus | 60.10 | 61.14 |  0.55  | 1.04 |
> | 4 | car | 85.82 | 87.42 |  4.42  | 1.60 |
> | 5 | construction_vehicle | 14.89 | 17.43 |  0.18  | 2.54 |
> | 6 | motorcycle | 38.61 | 41.16 |  0.05  | 2.55 |
> | 7 | pedestrian | 59.97 | 66.44 |  0.27  | 6.47 |
> | 8 | traffic_cone | 51.29 | 59.15 |  0.09  | 7.86 |
> | 9 | trailer | 53.88 | 54.48 |  0.58  | 0.60 |
> | 10 | truck | 62.08 | 63.10 |  1.83  | 1.02 |
> | 11 | driveable_surface | 94.82 | 93.35 |  37.58  | -1.47 |
> | 12 | flat_other | 51.77 | 54.37 |  1.02  | 2.60 |
> | 13 | sidewalk | 69.36 | 66.96 |  8.31  | -2.40 |
> | 14 | terrain | 67.53 | 71.67 |  8.34  | 4.14 |
> | 15 | manmade | 85.33 | 89.74 |  21.14  | 4.41 |
> | 16 | vegetation | 84.24 | 89.59 |  14.54  | 5.35 |
> | **Average** | | **59.42** | **62.07** | – | **2.65** |
>
> **Table R2.** Per-class improvements of pseudo-labels on unlabeled data from nuScenes-lidarseg with 1% labeled ratio.
>
> As the results demonstrate, several underrepresented categories (e.g., pedestrian, traffic_cone, person) show particularly large gains, highlighting the robustness of our refinement approach across different class frequencies.

---

> ### Author Response · Authors · 2025-11-20
> **Official Comment by Authors (Part 3/5)**
>
> ### **Experiments on an additional benchmark**
>
> Thank you for the suggestion. Expanding the evaluation beyond nuScenes-lidarseg and SemanticKITTI is certainly valuable for assessing how broadly the method applies.
>
> At the same time, the datasets you mentioned currently lack well established semi-supervised LiDAR semantic segmentation pipelines and reference baselines, which makes consistent and fair evaluation difficult. This is a practical limitation rather than a methodological choice on our side.
>
> To extend validation within feasible limits, we are running additional experiments on ScribbleKITTI, which provides a different supervision setting and helps us examine the behavior of the refinement mechanism with more sparse annotations.
>
> The table below summarizes the current results. For now, we have completed the 1% and 10 % labeled setting, respectively, and our method achieves the state-of-the-art performance in both settings.
>
> | **Method**      | **1%**   | **10%**  |
> | --------------- | -------- | -------- |
> | sup.            | 39.2     | 48.0     |
> | MT [1]         | 41.0     | 50.1     |
> | CBST [2]       | 41.5     | 50.6     |
> | CPS [3]         | 41.4     | 51.8     |
> | LaserMix [4]   | 44.2     | 53.7     |
> | IT2 [5]           | 47.9     | 56.7     |
> | FrustumMix [6]      | 45.6     | 55.7     |
> | LaserMix++ [7]      | 47.3     | 56.7     |
> | **RePL (Ours)** | **48.7** | **57.3** |
>
> **Table R3.**  Comparison of different semi-supervised learning methods on ScribbleKITTI while varying the ratio of labeled data.
>
> We are continuing experiments on other labeled ratios and will report the results as soon as they become available.  Additionally, we are preparing experiments on Waymo Open Dataset to broaden the evaluation scope, and will include any completed results.
>
> &nbsp;
>
>
> ### **Additional Qualitative Analysis on Failure Cases and Improvement on Long-Range Regions**
>
> Thank you for pointing this out.
>
> We agree that understanding where REPL is less effective is important for a comprehensive evaluation. To address this, we included additional qualitative results that focus explicitly on failure cases in Figure 4 of the revised paper and discussed them in a separate paragraph (line 527).
>
> Furthermore, we present in-depth visual comparisons in Figure 12 and Figure 13 that examine long-range scenarios; regions where our refinement approach demonstrates notable effectiveness in mitigating noise and correcting segmentation errors. Extended discussion of these results can be found in the appendix (line 856).
>
> We also examined class-wise performance changes in **Table R1** and **R2**, for both SemanticKITTI and nuScenes-lidarseg, respectively. The improvements are consistent for most object classes, but smaller gains appear in categories with few points or highly fragmented geometry, including pedestrians, traffic cones, and bicycles.
> These additions clarify the strengths and remaining limitations of REPL and provide a more complete picture of its behavior.
>
>
>
>
>
>
>
>
> &nbsp;
>
> &nbsp;
>
> [1] MT [Tarvainen & Valpola] Mean teachers are better role models: Weight-averaged consistency targets improve semi-supervised deep learning results. NeurIPS, 2017.
>
> [2] CBST [Zou, et al.] Unsupervised domain adaptation for semantic segmentation via class-balanced self-training. ECCV, 2018.
>
> [3] CPS [Chen, et al.] Semi-supervised semantic segmentation with cross pseudo supervision. CVPR, 2021.
>
> [4] LaserMix [Kong, et al.] Lasermix for semi-supervised lidar semantic segmentation. CVPR, 2023.
>
> [5] IT2 [Liu, et al.] Ittakestwo: Leveraging peer representations for semi-supervised lidar semantic segmentation. ECCV, 2024.
>
> [6] FRNet [Xu, et al.] FRNet: Frustum-Range Networks for Scalable LiDAR Segmentation. TIP, 2025.
>
> [7] LaserMix++ [Kong, et al.] Multi-Modal Data-Efficient 3D Scene Understanding for Autonomous Driving. TPAMI, 2025.

---

> > ### Author Response · Authors · 2025-11-30
> > **Additional Results on ScribbleKITTI with 20% Labeled Data**
> >
> > We have completed the requested experiments on ScribbleKITTI with 20% labeled data and present the comprehensive results below:
> >
> > | **Method** | **1%** | **10%** | **20%** | **Average** |
> > | --------------- | -------- | -------- | ------- | ----------- |
> > | sup. | 39.2 | 48.0 | 52.1 | 46.4 |
> > | MT [1] | 41.0 | 50.1 | 52.8 | 48.0 |
> > | CBST [2] | 41.5 | 50.6 | 53.3 | 48.5 |
> > | CPS [3] | 41.4 | 51.8 | 53.9 | 49.0 |
> > | LaserMix [4] | 44.2 | 53.7 | 55.1 | 51.0 |
> > | IT2 [5] | 47.9 | 56.7 | 57.5 | 54.0 |
> > | FrustumMix [6] | 45.6 | 55.7 | **58.2** | 53.2 |
> > | LaserMix++ [7] | 47.3 | 56.7 | 57.6 | 53.9 |
> > | **RePL (Ours)** | **48.7** | **57.3** | 57.6 | **54.5** |
> >
> > **Table R5**. Comparison of different semi-supervised learning methods on ScribbleKITTI while varying the ratio of labeled data.
> >
> > RePL achieves the highest average performance (54.5) across the evaluated settings, demonstrating comprehensive superiority over existing methods.
> >
> > We are currently conducting experiments with 50% labeled data and will update these results in this rebuttal and include them in the revised manuscript upon completion.
> >
> > &nbsp;
> >
> > &nbsp;
> >
> > [1] MT [Tarvainen & Valpola] Mean teachers are better role models: Weight-averaged consistency targets improve semi-supervised deep learning results. NeurIPS, 2017.
> >
> > [2] CBST [Zou, et al.] Unsupervised domain adaptation for semantic segmentation via class-balanced self-training. ECCV, 2018.
> >
> > [3] CPS [Chen, et al.] Semi-supervised semantic segmentation with cross pseudo supervision. CVPR, 2021.
> >
> > [4] LaserMix [Kong, et al.] Lasermix for semi-supervised lidar semantic segmentation. CVPR, 2023.
> >
> > [5] IT2 [Liu, et al.] Ittakestwo: Leveraging peer representations for semi-supervised lidar semantic segmentation. ECCV, 2024.
> >
> > [6] FRNet [Xu, et al.] FRNet: Frustum-Range Networks for Scalable LiDAR Segmentation. TIP, 2025.
> >
> > [7] LaserMix++ [Kong, et al.] Multi-Modal Data-Efficient 3D Scene Understanding for Autonomous Driving. TPAMI, 2025.

---

> ### Author Response · Authors · 2025-11-20
> **Official Comment by Authors (Part 4/5)**
>
> ### **Computational Cost & Efficiency and Practicality for Large-Scale Deployment**
>
> To quantify the additional overhead by the refiner, we measured the latency and memory usage during inference on the validation set using a single batch.
>
> As shown in the table below, the refiner adds approximately 0.25 seconds of latency and 396 MB of memory, while providing a substantial improvement of +9.1 mIoU from the supervised-only baseline. These results demonstrate that the added computational cost is moderate relative to the significant accuracy gains.
>
> | **Method**                  | **Latency(s)** | **Memory (MB)** | **mIoU**  |
> | --------------------------- | -------------- | --------------- | --------- |
> | Baseline (supervised-only)  | 0.43           | 1231            | 50.9      |
> | Baseline+Refiner (semi-sup) | 0.68           | 1627            | 60.0      |
> | Δ                           | +0.25s         | +396 MB         | +9.1 mIoU |
>
> **Table R4.** Computational cost analysis on nuScenes-lidarseg.
>
> We have described this results in the revision (Table 7, line 478)
>
> Regarding practical deployment at scale, the additional 0.25s latency and 396 MB memory remain within acceptable ranges for many real-world applications, particularly when considering the substantial +9.1 mIoU improvement in segmentation quality. For safety-critical applications such as autonomous driving, where accuracy is a major concern, this cost-performance trade-off is favorable. Moreover, optimizations such as model compression or efficient inference techniques can further reduce the overhead while maintaining most of the performance gains.
>
> &nbsp;
>
> ### **Performance Improvement on Long-tailed Classes**
>
> To assess how REPL behaves on rare and long tail categories, we conducted a class wise analysis on unlabeled sets of both SemanticKITTI and nuScenes-lidarseg under the 1% labeled setting in **Table R1** and **R2** for both datasets, respectively.
> In the table, we report the relative frequency of each class ("frequency. (%)") within the entire dataset.
>
> Many extremely rare classes in SemanticKITTI, including bicycle, motorcycle, other vehicle, person, other ground, trunk, and pole, show clear gains despite their very low occurrence rates, often below 0.1%. Several of these categories improve by more than 2%p IoU, and person and other-ground increase by almost four points. Only motorcyclist, which is nearly absent in the dataset, fluctuates slightly. In contrast, dominant classes such as road, building, and vegetation still improve but often by a smaller margin.
>
> Overall, the refinements do not correlate strongly with class frequency, and rare classes benefit in a comparable way to frequent ones.
>
> The same observation holds for nuScenes-lidarseg. Long-tailed categories such as bicycle, construction vehicle, motorcycle, pedestrian, and traffic cone all improve, with pedestrian and traffic cone showing the largest gains. High frequency classes like manmade and vegetation also see notable improvements, while a few very large area classes such as driveable surface and sidewalk fluctuate slightly.
>
> *The overall trend is that REPL helps a broad range of classes regardless of their frequency, rather than concentrating its improvements on only the dominant categories.*

---

> ### Author Response · Authors · 2025-11-20
> **Official Comment by Authors (Part 5/5)**
>
> ### **Room for Improvement with Learned Error Detectors**
>
> Thank you for the question.
>
> We agree that a learned error detector has the potential to further improve the quality of the estimated error map. In the current framework, we intentionally adopt a very simple agreement based heuristic to keep the focus on the refinement mechanism itself. Despite its simplicity, this strategy is already sufficient to reach strong performance.
>
> At the same time, Table 4 of the main paper suggests that there is substantial room for additional gains if the error detection signal becomes more accurate (line 466). This indicates that our current choice is not close to the upper bound, and advanced error  detectors will likely provide better guidance for the refiner. Exploring such detectors is a promising direction, and we view it as a natural extension of the framework rather than a limitation of the current design.
>
> &nbsp;
>
> ### **Extension to Temporal Settings**
>
> Thank you for bringing up this idea.
>
> Extending the refinement process to sequential or spatio temporal LiDAR settings is indeed a promising direction. Temporal consistency can provide additional cues for identifying unreliable regions and may further stabilize the reconstruction based refinement.
>
> Our current work focuses on single frame 3D LiDAR, but the framework is compatible with incorporating temporal structure, such as neighborhood aggregation across consecutive sweeps or motion aware masking strategies. We plan to explore these extensions as future work, and we greatly appreciate the suggestion that aligns well with how our method could naturally evolve.

---

> ### Author Response · Authors · 2025-11-26
> **Gentle Reminder**
>
> Thank you once again for your thoughtful review. Your feedback has been incredibly helpful in improving our work.
>
> As the discussion period is nearing its end, we would be grateful if you could review the modifications and additional analysis we have made in response to your concerns. We hope these revisions address the points you raised and would appreciate your feedback at your convenience.
>
> We sincerely appreciate your valuable time and effort in reviewing our work.

---

### Official Review · Reviewer_2qeu · 2025-11-03

**Soundness:** 3
**Presentation:** 3
**Contribution:** 2
**Rating:** 4
**Confidence:** 4

**Summary:**

This paper presents REPL, a new framework for semi-supervised learning in LiDAR semantic segmentation. The core contribution is a shift from adjusting pseudo-label usage to actively improving their quality. The method identifies unreliable pseudo-labels using a heuristic and then corrects them using a masked reconstruction mechanism. The authors demonstrate state-of-the-art performance on the nuScenes-lidarseg and SemanticKITTI benchmarks.

**Strengths:**

1. The paper is well-written, logically structured, and easy to follow.

2. The analysis of the trade-off between error correction ($q_j$) and error introduction demonstrates the refiner's effectiveness .

3. The paper provides a rich set of ablation studies (e.g., Table 2-5) that thoroughly validate the contributions of the individual components, such as the refiner losses and the masking strategies.

**Weaknesses:**

1. The paper claims that pseudo-label quality improves during training due to the refiner. However, in teacher–student frameworks, the teacher naturally improves over time thanks to EMA updates. It is thus hard to isolate how much of the improvement is due to the refiner versus the inherent dynamics of self-training. A comparison between EMA-only and EMA+refiner over time would strengthen the claim.

2. The refiner is based on masked reconstruction, which is generative by nature. It is unclear whether the gains stem from genuine pseudo-label correction or simply from the introduction of stronger feature representations via MAE. Ablations that replace MAE with another feature extractor (e.g., contrastive learning backbone) or apply MAE without pseudo-label refinement would help clarify this distinction.

3. The 'unreliable voxel identification' heuristic relies on several parameters (e.g., confidence percentile $\kappa$). This raises a question about potential hyperparameter sensitivity. The paper would be strengthened by an analysis of how performance varies with these parameters, as it is currently challenging to gauge the tuning effort that might be required for new datasets.

4. The framework introduces a third network (the refiner), which presumably adds computational and memory overhead. The paper does not currently include an analysis of training time or memory usage relative to baseline methods. Including this would help readers better evaluate the practical performance-vs-efficiency trade-off.

5. The evaluation could be further strengthened by including additional benchmarks. For instance, key prior work like LaserMix also reported results on the ScribbleKITTI dataset. An evaluation on this would provide a more comprehensive picture of the method's robustness.

**Questions:**

na

---

> ### Author Response · Authors · 2025-11-20
> **Official Comment by Authors (Part 1/3)**
>
> Thank you for the insightful and constructive comment! Below, we provide detailed responses to address your concerns.
>
> &nbsp;
>
> ### **A comparison between EMA-teacher only and EMA-teacher with refiner over time**
>
> We appreciate your comment and agree that it is essential to disentangle improvements stemming from the EMA teacher’s natural update dynamics from those introduced by our refiner.
> To examine this, we compare the pseudo-label quality on the unlabeled set of nuScenes-lidarseg produced by (i) the EMA teacher alone and (ii) the EMA teacher followed by our refiner. **The resulting trends are shown in Figure 6 and described in Section A.4 in the appendix of the revised paper.**
>
> Across all label ratios in nuScenes-lidarseg (1%, 10%, 20%, 50%), we observe a consistent pattern: the refiner provides additional improvements on top of the EMA teacher throughout training. This suggests that the gains are not solely due to EMA updates but also reflect the contribution of the refinement process.
>
> We also highlight this effect in the main text (Figure 5) and in the appendix (SemanticKITTI results in Figure 7), where the same trend appears. Collectively, these comparisons indicate that the refiner complements the teacher rather than merely mirroring its natural progression.
>
> &nbsp;
>
> ### **Clarifying the Role of MAE: No Pretrained or Stronger Representations Used**
>
> Thank you for this comment. We understand that the use of masked reconstruction may give the impression that our refiner benefits from MAE-style strong feature representations. **However, this is not the case in our setting.**
>
> Our method does not use an MAE-pretrained backbone, nor does it adopt any externally pretrained feature extractor. The refiner shares the **same randomly-initialized Cylinder3D architecture as both the student and the EMA teacher.**
>
> The only MAE-related component is the training objective, where the refiner learns through a masking-and-reconstruction loss, but the underlying feature extractor is not replaced, strengthened, or pre-initialized by MAE.
>
> This means the outstanding performance of our method does not stem from importing a stronger representation. Instead, it arises from the refiner’s ability to adjust and correct pseudo-labels through its masked reconstruction mechanism, while operating within the same representational capacity as the teacher and student.

---

> ### Author Response · Authors · 2025-11-20
> **Official Comment by Authors (Part 2/3)**
>
> ### **On the Sensitivity of the Unreliable-Voxel Identification Parameters**
>
> Thanks for the great suggestions!
>
> We first gently clarify that the unreliable pseudo-label identification process depends on only one essential hyper-parameter: the confidence percentile $\kappa$, which determines the threshold for identifying unreliable pseudo-labels. We analyzed the sensitivity of our method on the nuScenes-lidarseg validation set, trained with 1% labelled ratio. As shown in the table below, the performance exhibits moderate sensitivity to $\kappa$.
>
> | κ    | mIoU |
> | ---- | ---- |
> | 0.2  | 55.1 |
> | 0.4  | 60.0 |
> | 0.6  | 58.4 |
>
> **Table R1.**  Sensitivity analysis of a hyper-parameter κ.
>
> We have described these results in the revision (Table 6, line 483).
>
>
> To mitigate this sensitivity, we recommend using $\kappa\in[0.3,0.5]$ as a starting range. Values below 0.2 should be avoided as they identify too few unreliable samples for effective refinement, while values above 0.6 may incorrectly flag too many confident predictions. If validation data is available, a quick test of $ \kappa \in $ { $ 0.3, 0.4, 0.5 $ } requires minimal cost and helps find the dataset-specific optimum. Across our experiments, $\kappa = 0.4$ performed reliably.
>
> &nbsp;
>
> ### **Computational Cost & Efficiency**
>
> Thanks for the constructive comment.
>
> To quantify the additional overhead by the refiner, we measured the latency and memory usage during inference on the validation set using a single batch.
>
> As shown in the table below, the refiner adds approximately 0.25 seconds of latency and 396 MB of memory, while providing a substantial improvement of +9.1 mIoU from the supervised-only baseline. These results demonstrate that the added computational cost is moderate relative to the significant accuracy gain.
>
> | **Method**                   | **Latency(s)** | **Memory (MB)** | **mIoU**  |
> | ---------------------------- | -------------- | --------------- | --------- |
> | Baseline (supervised only)   | 0.43           | 1231            | 50.9      |
> | Baseline+Refiner (semi-sup.) | 0.68           | 1627            | 60.0      |
> | Δ                            | +0.25s         | +396 MB         | +9.1 mIoU |
>
> **Table R2.** Computational cost analysis on nuScenes-lidarseg.
>
> We have described these results in the revision (Table 7, line 478)

---

> ### Author Response · Authors · 2025-11-20
> **Official Comment by Authors (Part 3/3)**
>
> ### **Additional Experiments on ScribbleKITTI**
>
> What a great suggestion!
>
> To provide a more comprehensive evaluation, we additionally ran experiments on ScribbleKITTI.
> The table below summarizes the current results. For now, we have completed experiments in the 1% and 10 % labeled setting, respectively, and our method achieves the state-of-the-art performance among existing approaches for both settings.
>
> | **Method**      | **1%**   | **10%**  |
> | --------------- | -------- | -------- |
> | sup-only            | 39.2     | 48.0     |
> | MT [1]         | 41.0     | 50.1     |
> | CBST [2]       | 41.5     | 50.6     |
> | CPS [3]         | 41.4     | 51.8     |
> | LaserMix [4]   | 44.2     | 53.7     |
> | IT2 [5]            | 47.9     | 56.7     |
> | FrustumMix [6]      | 45.6     | 55.7     |
> | LaserMix++  [7]    | 47.3     | 56.7     |
> | **RePL (Ours)** | **48.7** | **57.3** |
>
> **Table R3**. Comparison of different semi-supervised learning methods on ScribbleKITTI while varying the ratio of labeled data.
>
> We are continuing experiments in other settings and will report the results as soon as they become available.
>
> &nbsp;
>
> &nbsp;
>
> [1] MT [Tarvainen & Valpola] Mean teachers are better role models: Weight-averaged consistency targets improve semi-supervised deep learning results. NeurIPS, 2017.
>
> [2] CBST [Zou, et al.] Unsupervised domain adaptation for semantic segmentation via class-balanced self-training. ECCV, 2018.
>
> [3] CPS [Chen, et al.] Semi-supervised semantic segmentation with cross pseudo supervision. CVPR, 2021.
>
> [4] LaserMix [Kong, et al.] Lasermix for semi-supervised lidar semantic segmentation. CVPR, 2023.
>
> [5] IT2 [Liu, et al.] Ittakestwo: Leveraging peer representations for semi-supervised lidar semantic segmentation. ECCV, 2024.
>
> [6] FRNet [Xu, et al.] FRNet: Frustum-Range Networks for Scalable LiDAR Segmentation. TIP, 2025.
>
> [7] LaserMix++ [Kong, et al.] Multi-Modal Data-Efficient 3D Scene Understanding for Autonomous Driving. TPAMI, 2025.

---

> > ### Author Response · Authors · 2025-11-30
> > **Additional Results on ScribbleKITTI with 20% Labeled Data**
> >
> > We have completed the requested experiments on ScribbleKITTI with 20% labeled data and present the comprehensive results below:
> >
> > | **Method** | **1%** | **10%** | **20%** | **Average** |
> > | --------------- | -------- | -------- | ------- | ----------- |
> > | sup. | 39.2 | 48.0 | 52.1 | 46.4 |
> > | MT [1] | 41.0 | 50.1 | 52.8 | 48.0 |
> > | CBST [2] | 41.5 | 50.6 | 53.3 | 48.5 |
> > | CPS [3] | 41.4 | 51.8 | 53.9 | 49.0 |
> > | LaserMix [4] | 44.2 | 53.7 | 55.1 | 51.0 |
> > | IT2 [5] | 47.9 | 56.7 | 57.5 | 54.0 |
> > | FrustumMix [6] | 45.6 | 55.7 | **58.2** | 53.2 |
> > | LaserMix++ [7] | 47.3 | 56.7 | 57.6 | 53.9 |
> > | **RePL (Ours)** | **48.7** | **57.3** | 57.6 | **54.5** |
> >
> > **Table R4**. Comparison of different semi-supervised learning methods on ScribbleKITTI while varying the ratio of labeled data.
> >
> > RePL achieves the highest average performance (54.5) across the evaluated settings, demonstrating comprehensive superiority over existing methods.
> >
> > We are currently conducting experiments with 50% labeled data and will update these results in this rebuttal and include them in the revised manuscript upon completion.
> >
> > &nbsp;
> >
> > &nbsp;
> >
> > [1] MT [Tarvainen & Valpola] Mean teachers are better role models: Weight-averaged consistency targets improve semi-supervised deep learning results. NeurIPS, 2017.
> >
> > [2] CBST [Zou, et al.] Unsupervised domain adaptation for semantic segmentation via class-balanced self-training. ECCV, 2018.
> >
> > [3] CPS [Chen, et al.] Semi-supervised semantic segmentation with cross pseudo supervision. CVPR, 2021.
> >
> > [4] LaserMix [Kong, et al.] Lasermix for semi-supervised lidar semantic segmentation. CVPR, 2023.
> >
> > [5] IT2 [Liu, et al.] Ittakestwo: Leveraging peer representations for semi-supervised lidar semantic segmentation. ECCV, 2024.
> >
> > [6] FRNet [Xu, et al.] FRNet: Frustum-Range Networks for Scalable LiDAR Segmentation. TIP, 2025.
> >
> > [7] LaserMix++ [Kong, et al.] Multi-Modal Data-Efficient 3D Scene Understanding for Autonomous Driving. TPAMI, 2025.

---

> ### Author Response · Authors · 2025-11-26
> **Gentle Reminder**
>
> Thank you once again for your thoughtful review. Your feedback has been incredibly helpful in improving our work.
>
> As the discussion period is nearing its end, we would be grateful if you could review the modifications and additional analysis we have made in response to your concerns. We hope these revisions address the points you raised and would appreciate your feedback at your convenience.
>
> We sincerely appreciate your valuable time and effort in reviewing our work.

---

> > ### Comment · Reviewer_2qeu · 2025-11-28
> > **Post rebuttal**
> >
> > The authors present a clear and thorough rebuttal, directly addressing each concern with new experiments, detailed analyses, and quantitative evidence. Their responses convincingly clarify the role of the refiner, the behavior of pseudo-label dynamics, and the computational implications.
> >
> > While the rebuttal itself is excellent and resolves the raised questions convincingly, the conceptual novelty of the method still feels limited, as it is largely built upon established components in semi-supervised learning. Even so, given the method’s demonstrated effectiveness and practical utility, I am inclined to update my score accordingly.

---

> > > ### Author Response · Authors · 2025-11-28
> > > **Thank you for the positive evaluation of our rebuttal**
> > >
> > > We sincerely appreciate your thorough review and the positive assessment that our rebuttal "is excellent and resolves the raised questions convincingly."
> > >
> > > We also appreciate your willingness to reconsider the score in light of the clarified results and analyses.
> > >
> > > Regarding the novelty concern, we would like to respectfully clarify our contributions. The individual components we use are indeed established techniques, and we do not claim them as standalone contributions. Our contribution is in introducing, to the best of our knowledge, the first refinement framework specifically designed for 3D LiDAR pseudo-labels, demonstrating that masked reconstruction is a powerful tool in this context.  We also would claim that our theoretical analyses are a notable contribution, offering strong impact on label-efficient LiDAR segmentation.
> > >
> > > We below discuss our contributions in more detail.
> > >
> > > - We clarify that the agreement-based unreliable voxel identification **is made simple on purpose to better demonstrate the power of our refinement method, and is not claimed as a core contribution**. It functions as a minimal mechanism to expose the refinement process to informative signals, and any advanced unreliable voxel detection methods can further improve the performance as demonstrated in Table 4 of the main paper.
> > >
> > > - One of the main contributions lies in **using masked reconstruction as a refiner for correcting 3D LiDAR pseudo-labels**. Masked reconstruction is commonly used as a representation learning objective, but applying it as a label correction operator in sparse voxelized LiDAR requires dedicated designs specific to this domain. **We believe that bringing this mechanism into LiDAR segmentation and demonstrating its outstanding performance is clearly a contribution of our work.**
> > >
> > > - To our knowledge, **our work is the first to formulate and systematize pseudo-label refinement for 3D LiDAR segmentation.** Prior methods rely on consistency between teacher and student without an explicit refinement module. Our formulation defines unreliable region identification, reconstruction-based correction, and the separation of roles among components, providing a clear framework for pseudo-label refinement in LiDAR segmentation.
> > >
> > > - We provide theoretical analyses explaining why reconstruction-driven refinement can correct label noise under realistic conditions (see Sec. 3.3). These analyses support our idea theoretically and distinguish our work from a simple aggregation of existing ideas.
> > >
> > > &nbsp;
> > >
> > > We believe that introducing masked reconstruction to LiDAR pseudo-label refinement, supported by theoretical analysis and strong empirical results across multiple benchmarks, represents a meaningful contribution to the field.
> > >
> > >
> > > Once again, thank you for your constructive feedback throughout the review process.

---

### Author Response · Authors · 2025-11-21
**Common Response to All Reviewers**

We sincerely thank all reviewers for their thorough evaluations and constructive feedback. We are particularly grateful for the recognition of our novel refinement approach that directly improves pseudo-label quality at its source (all), the comprehensive experimental validation including ablation studies and sensitivity analyses (all), the principled theoretical analysis that is relatively rare in semi-supervised segmentation (yQAk), the insightful analysis of the error correction-introduction trade-off (2qeu), and the alignment of our masked reconstruction approach with recent self-supervised learning advances (X4n8). Before addressing individual concerns, we would like to briefly outline the key points of our rebuttal:

**Additional Analysis**

* Analysis on computational cost and efficiency of the refiner.
* Analysis on hyper-parameter sensitivity of the confidence percentile $ \kappa $.
* Analysis on pseudo-label quality improvement throughout training.
* Enhanced theoretical explanation of Proposition 2.

**Extended Evaluation**

* Additional evaluation on ScribbleKITTI benchmark.
* Additional per-class performance analysis on rare and frequent classes.
* Additional failure case analysis and long-range scenario visualization.
* Broader method comparison including FRNet [1], LaserMix++ [2], Seal [3], SuperFlow [4], and SLidR [5].

Please note that we are continuing with additional experiments and will update our progress and responses as soon as they are completed.

&nbsp;

&nbsp;

[1] FRNet [Xu, et al.] FRNet: Frustum-Range Networks for Scalable LiDAR Segmentation. TIP, 2025.

[2] LaserMix++ [Kong, et al.] Multi-Modal Data-Efficient 3D Scene Understanding for Autonomous Driving. TPAMI, 2025.

[3] Seal [Liu, et al.] Segment Any Point Cloud Sequences by Distilling Vision Foundation Models. NeurIPS, 2023.

[4] SuperFlow [Xu, et al.] 4D Contrastive Superflows are Dense 3D Representation Learners. ECCV, 2024.

[5] SLidR [Sautier, et al.] Image-to-Lidar Self-Supervised Distillation for Autonomous Driving Data. CVPR, 2022.

---

### Author Response · Authors · 2025-12-03
**Summary on Rebuttal Progress**

We sincerely thank all reviewers for their constructive feedback and engagement throughout the discussion period. We are pleased to report significant progress during the rebuttal phase:

&nbsp;

### **Reviewer Responses**

- **Reviewer X4n8** has maintained their positive rating and noted that "most of the concerns raised from my side have been well addressed," recommending acceptance.

- **Reviewer 2qeu**, the only negative reviewer whose original score was 4, has indicated their willingness to reconsider their score, stating that our rebuttal "is excellent and resolves the raised questions convincingly."

- **Reviewer yQAk**, while not participating in the discussion period, championed our work because of its theoretical background, strong performance, and clarity of presentation. They raised concerns about potential brittleness in challenging cases, a narrow evaluation scope, lack of failure cases illustrated, and computational costs. We throughly addressed these by demonstrating robustness via per-class analysis on rare categories, additional experiments on ScribbleKITTI, providing failure cases with discussions, and quantified computational costs.

&nbsp;

### **Key Improvements to Our Paper**

In response to the reviewer feedback, we have substantially improved our paper as follows:

- Extended experimental validation on the ScribbleKITTI benchmark across multiple supervision ratios (Reviewers 2qeu, yQAk)
- Comprehensive per-class analysis demonstrating benefits on rare and challenging categories (Reviewers yQAk, X4n8)
- Computational cost analysis quantifying the practical trade-offs (Reviewers 2qeu, X4n8)
- Temporal analysis of pseudo-label quality improvement with detailed visualizations (Reviewer 2qeu)
- Sensitivity analysis on key hyperparameters with practical guidelines (Reviewer 2qeu)
- Enhanced theoretical explanations and broader method comparisons (Reviewer X4n8)
- Transparent failure case analysis and limitation discussions (Reviewer yQAk)

&nbsp;

These additions have significantly strengthened the rigor and completeness of our work by demonstrating generalization across benchmarks, providing practical guidance through detailed analyses, ensuring transparency, and clarifying theoretical contributions for broader accessibility. We sincerely appreciate the reviewers for their constructive comments that enable this revision, and remain available to address any remaining questions or concerns!

---

### Meta-Review · Area_Chair_6KsD · 2026-01-02

**Summary:**

The paper initially received one negative and two positive ratings. The concerns are mostly about 1) technical novelty, 2) effectiveness of the proposed components, 3) more analysis, e.g., parameter sensitivity, computational efficiency, performance on rare classes, failure cases, 4) more experiments, e.g., dataset, baseline comparisons.

**Reviewer Concerns:**

The authors have provided responses in the rebuttal to answer initial concerns from the reviewers. The AC took a close look at the paper, reviews, and the rebuttal. After the rebuttal, the AC finds that some questions are addressed with more experiments, e.g., analysis on parameter sensitivity, computational efficiency, and results on ScribbleKITTI. However, there are still remaining concerns that are not addressed well. To be specific, reviewer 2qeu is still concerned about the technical novelty (similar concerns shared by reviewer yQAk who did not comment after rebuttal), while reviewer X4n8 also agrees with this limitation but focuses more on how the authors integrate all the existing components. The AC considers both sides' opinions, while still finding unconvincing results shared by reviewer 2qeu and yQAk, particularly on the limited performance gains compared with SOTA like LaserMix++ and rare classes (i.e., there is only limited improvement on pseudo-labels but not final results). Considering the limited technical insight, novelty, and contributions, the AC agrees with the reviewers' overall feedback and hence recommends the rejection rating, while encouraging the authors to improve the manuscript based on reviewers' feedback and resubmit the work to another venue.

**Reviewer Scores:**

Reviewer 2qeu and X4n8 mentioned to retain the original rating, while another reviewer did not fully participate in the discussion.

---

### Decision · Program_Chairs · 2026-01-26

Reject